# Point-of-care early infant HIV diagnosis at birth in a pragmatic cluster-randomized trial in Mozambique and Tanzania: A comparative cost and cost-effectiveness study

Kira Elsbernd[1,2,3*], Issa Sabi[4], Ilesh V. Jani[5], Chishamiso Mudenyanga[6],
Siriel Boniface[4], Arlete Mahumane[5], Joaquim Lequechane[5], Falume Chale[5],
Bindiya Meggi[5], Kassia Pereira[5], Raphael Edom[4], Anange F. Lwilla[4],
W. Chris Buck[7], Nyanda Elias Ntinyinya[4], Michael Hoelscher[1,3,8,9],
Till Baernighausen[10,11], Arne Kroidl[1,3], Stefan Kohler[10,11,12],
the LIFE Study Consortium¶

1 Institute of Infectious Diseases and Tropical Medicine, LMU University Hospital, LMU Munich, Munich, Germany, 2 Institute for Medical Information Processing, Biometry, and Epidemiology, LMU Munich, Munich, Germany, 3 German Center for Infection Research (DZIF), partner site Munich, Munich, Germany, 4 National Institute for Medical Research, Mbeya, Tanzania, 5 Instituto Nacional de Saúde, Maputo, Mozambique, 6 Clinton Health Access Initiative, Maputo, Mozambique, 7 David Geffen School of Medicine, University of California Los Angeles, Los Angeles, California, United States of America, 8 Immunology, Infection and Pandemic Research, Fraunhofer Institute for Translational Medicine and Pharmacology ITMP, Munich, Germany, 9 Unit Global Health, Helmholtz Center Munich, German Research Center for Environmental Health (HMGU), Neuherberg, Germany, 10 Heidelberg Institute of Global Health, Faculty of Medicine and University Hospital, Heidelberg University, Heidelberg, Germany, 11 German Center for Infection Research (DZIF), partner site Heidelberg, Heidelberg, Germany, 12 Institute of Social Medicine, Epidemiology and Health Economics, Charité – Universitätsmedizin Berlin, Corporate Member of Freie Universität Berlin and Humboldt-Universität zu Berlin, Berlin, Germany

¶ Membership of the LIFE Study Consortium is listed in the Acknowledgments.
* elsbernd.kira@med.uni-muenchen.de

## Abstract

### Background

Timely access to early infant diagnosis (EID) is crucial for newborns with HIV, as late diagnosis can delay lifesaving antiretroviral treatment (ART). We assessed the comparative cost and cost-effectiveness of integrating point-of-care EID at birth into routine care in primary healthcare settings.

### Methods and findings

This pre-specified secondary analysis was nested in the cluster-randomized LIFE study conducted at 28 primary healthcare facilities in Mozambique and Tanzania from October 2019 to September 2021. We estimated the health system cost of point-of-care birth plus 4–8-week HIV testing (very early infant diagnosis; VEID) compared to standard-of-care (SoC) testing at 4–8 weeks only, both with immediate ART initiation. We assessed the cost-effectiveness of VEID relative to SoC with respect to ART initiation within one week of life using Bayesian hierarchical models. As this is

**Data availability statement:** Cost data used in this study are available in the manuscript and supplementary material. Access to individual-level health outcome data, which contain potentially identifying and sensitive patient information, can be requested by contacting the data management unit at the Institute of Infectious Diseases and Tropical Medicine, LMU University Hospital [Alberto.Beyersdorff@med.uni-muenchen.de]. Code used in the analysis is available from Zenodo [https://doi.org/10.5281/zenodo.14959301].

**Funding:** This study was supported by the European and Developing Countries Clinical Trials Partnership (EDCTP; www.edctp.org) Grant No. RIA2016MC-1615 to IS, IJ, and AK as principal investigators, UNITAID (www.unitaid.org) Grant No. UCPOC2B to CM as principal investigator, and the German Center for Infection Research (DZIF; www.dzif.de) Grant No. TTU 04.708 to AK as principal investigator and Grant No. TTU 04.918 to SK, TB, AK, and KE. KE, MH, AK, TB, and SK are affiliated researchers at DZIF. The funders had no other role in study design, data collection and analysis, decision to publish, or preparation of the manuscript.

**Competing interests:** I have read the journal's policy and the authors of this manuscript have the following competing interests: TB is editor-in-chief of PLOS Medicine. The other authors have declared that no competing interests exist.

**Abbreviations:** ART, lifesaving antiretroviral treatment; CI, confidence interval; CrI, credible interval; EID, early infant diagnosis; ICERs, incremental cost-effectiveness ratios; IQR, interquartile range; LPV/r, lopinavir/ritonavir; NVP, nevirapine; PEPFAR, President's Emergency Plan for AIDS Relief; PoC, point-of-care; SoC, standard-of-care; VEID, very early infant diagnosis; WHO, World Health Organization; WTP, willingness-to-pay.

an intermediate outcome, incremental cost-effectiveness ratios (ICERs) cannot be directly compared to available life-year-based cost-effectiveness thresholds. To contextualize results, we derived the minimum life-years gained per early ART initiation required for VEID to meet standard thresholds in a break-even analysis.

VEID was associated with a higher cost and resulted in earlier ART initiation than SoC in both countries. In Mozambique, VEID increased the proportion of infants initiating ART within one week of life by 90.0 (95% CrI [67.5, 98.5]) percentage points at an incremental cost of $2,632 (95% CrI [$2,249, $3,062]) per infant with HIV. In Tanzania, VEID increased early ART initiation by 59.9 (95% CrI [20.9, 89.5]) percentage points at an incremental cost of $6,263 (95% CrI [$5,394, $7,243]) per infant with HIV. The ICER was $2,924 and $10,458 in Mozambique and Tanzania, respectively and was sensitive to intrauterine transmission rate. These findings were limited by the lack of long-term health outcome data and reliance on an intermediate outcome. Based on the break-even analysis, we estimated that VEID would need to yield 6–32 life-years gained per additional early ART initiation to meet standard thresholds.

## Conclusions

Adding birth testing improved early ART initiation but was unlikely to be cost-effective relative to standard thresholds given current prices, vertical transmission rates, and knowledge of long-term health benefits. Cost-effectiveness could be achieved at current costs if early ART translates to substantial long-term health benefits or if targeted to infants at high risk of vertical transmission.

## Author summary
### Why was this study done?

- Newborns who acquire HIV before birth are at high risk of illness and death unless they start treatment quickly.

- Testing for HIV at birth with rapid, same-day point-of-care (PoC) tests could identify these newborns sooner than the usual test at 4–8 weeks, allowing earlier treatment.

- Adding PoC testing at birth requires extra resources, and it is unclear whether the health gains justify the higher costs in low-resource countries.

### What did the researchers do and find?

- We evaluated PoC birth testing plus the routine 4–8 weeks test (called VEID) against the standard approach of testing only at 4–8 weeks (called SoC) within a pragmatic cluster-randomized trial at 28 primary healthcare facilities in Mozambique and Tanzania between 2019 and 2021.

- Infants born in VEID sites more often received PoC testing, more often received treatment, and started treatment earlier than in SoC sites.

- To be cost-effective, VEID would need to produce large long-term health gains.

## What do these findings mean?

- Adding birth PoC testing can speed up lifesaving treatment for infants with HIV, but universal roll-out in settings with low HIV transmission is unlikely to be cost-effective without targeted use, lower test prices, sharing of testing resources across programs, or large long-term benefits.

- A limitation of our study is that infants were followed only for the first few months of life, meaning we could not directly measure the long-term health benefits of starting treatment earlier. Instead, we estimated the minimum survival benefit required (6–32 additional years of life) for VEID to be considered cost-effective at current prices.

## Introduction

Timely access to HIV early infant diagnosis (EID) could improve the health outcomes of the 1.3 million children born to mothers living with HIV each year globally [1]. In 2023, approximately 120,000 of these children acquired HIV through gestation, birth, or breastfeeding. Early diagnosis is required for early antiretroviral treatment (ART) initiation, which is especially critical for neonates acquiring HIV in-utero, half of whom die before two years of age without treatment [2]. The World Health Organization (WHO) currently recommends that all infants born to mothers living with HIV receive an EID test by two months of life and all infants diagnosed with HIV immediately initiate ART [3]. Late diagnostic testing and frequent loss to retention after birth cause delays in access to ART [4], often past a peak in HIV-related mortality reported at 2–3 months of age [5].

Same-day point-of-care (PoC) EID has improved access and decreased time to treatment initiation by streamlining EID and linkage to care processes [6–9] and is cost-effective compared to laboratory-based testing [10]. Yet in 2023, only 67% of infants exposed to HIV were tested in the first two months of life and only 57% of children 0–14 years living with HIV were on ART [1]. Testing at birth, in addition to the standard 4–8 weeks of age, offers the possibility to identify infants acquiring HIV in-utero earlier. Immediate ART in the first weeks of life has the potential to reduce early mortality and morbidity, prevent or lessen the development of long-lasting viral reservoirs, and improve viral control [11–15].

EID programs will likely need to consider same-day test-and-treat in the first week of life to reduce persistently high early HIV-related mortality among infants. PoC EID is now widely available, but few sub-Saharan African countries have implemented birth testing into routine practice, partially due to uncertainty around costs and cost-effectiveness. A modelling study for South Africa suggests that adding birth testing to standard EID schedules is cost-effective [16]. Cost-effectiveness analyses informed by pragmatic trials of PoC EID at birth could provide additional support for program-level implementation, especially from resource-poor, rural, and peri-urban settings where effective EID programs are most needed.

This study was conducted within a pragmatic trial of PoC EID at birth in public primary healthcare facilities in Mozambique and Tanzania, where high HIV prevalence and persistent vertical transmission [1] suggest that adding birth test-and-treat could have a meaningful impact on HIV-related early infant mortality and morbidity in these settings. The objective of the study was to estimate the cost and cost-effectiveness of offering PoC EID and immediate ART to infants exposed to HIV under routine conditions at birth plus 4–8 weeks of age (very early infant diagnosis; VEID) compared with the standard-of-care (SoC) at 4–8 weeks of age only. Cost-effectiveness was evaluated with respect to early ART initiation and EID uptake, which are intermediate outcomes on the causal pathway to reduced early infant mortality [14].

## Methods

### Study setting

This trial-based cost and cost-effectiveness analysis was nested in the LIFE study (NCT04032522), a pragmatic cluster-randomized trial which took place at 28 primary healthcare facilities in the Sofala and Manica provinces of Mozambique and the Mbeya and Songwe regions of Tanzania (7 sites per country per arm). The LIFE study enrolled 6,602 infants born to women living with HIV from October 2019 to September 2021: 3,294 in VEID sites and 3,308 in SoC sites. Among 125 infants diagnosed with HIV until 12 weeks of age, the study demonstrated a clinically relevant but not significant reduction in mortality up to 6 months of age with VEID. A total of 65 infants (52.0%) were diagnosed with HIV at birth. Vertical transmission was 1.89% (95% confidence interval [CI] [1.58, 2.25]) overall with 86.4% of infants (108) from Mozambique and 13.6% (17) from Tanzania [15].

### Ethical considerations

Ethical approvals for the LIFE study were obtained from the Comité Institucional de Bioética para a Saúde of the Instituto Nacional de Saúde and Comité Nacional de Bioética em Saúde (Ref No. 509/CNBS/20) in Mozambique, the Mbeya Medical Research and Ethics Committee (Ref No. SZEC-2439/R.E/V. 1/90) and the Medical Research Coordinating Committee of the National Institute for Medical Research (Ref No. NIMR/HQ/R.8a/Vol. IX/3071) in Tanzania, and the Ethics Committee of the Ludwig Maximilians University Hospital (Ref No. 19–441) in Germany. All study participants provided written informed consent for themselves and their infants.

### Testing and treatment procedures

Half of the sites implemented PoC EID within 72 hours of birth and at 4–8 weeks of age (VEID) and the other half at 4–8 weeks of age only (SoC) (Fig 1). Infants with negative or unknown HIV status at birth started post-natal prophylaxis and received PoC EID at follow-up visits at 4–8 and 12 weeks of age (plus 4-week window period). HIV–positive results were confirmed by a second PoC EID test or, if a valid result could not be obtained on site, HIV-DNA performed at a central laboratory from dried blood spots. Infants diagnosed with HIV were immediately started on ART. Dried blood spots were collected for all infants at SoC sites at birth for retrospective analysis of HIV status at birth and in the case of death or loss to follow-up with unknown HIV status.

Following routine guidelines, all infants exposed to HIV in Mozambique were given enhanced post-natal prophylaxis with zidovudine (AZT) syrup for 6 weeks plus nevirapine (NVP) syrup for 12 weeks [17]. In Tanzania, only high-risk infants (i.e., mother newly diagnosed with HIV, not on ART or on ART for less than four weeks, or with high viral load in the last four weeks), were given enhanced post-natal prophylaxis. Standard post-natal prophylaxis with NVP syrup for 6 weeks was given to low-risk infants [18]. ART for infants with HIV aged 0–4 weeks and at least 2 kg consisted of AZT, lamivudine (3TC), and NVP syrups. At 4 weeks of age and at least 3 kg, infants were given abacavir (ABC)/3TC dispersible tablets and lopinavir/ritonavir (LPV/r) granules. Nurses were trained in sample collection and use of the PoC analyzers, performed all PoC testing and pre- and post-test counseling, and initiated ART with physician support.

### Testing platforms

The Abbott mPIMA HIV-1/2 Detect and Cepheid Xpert HIV-1 Qual were used for PoC EID testing in Mozambique and Tanzania, respectively, reflecting preexisting local HIV program preferences. The mPIMA analyzer is also approved for HIV viral load and can be procured with an external battery to bridge power outages. The Xpert analyzer can run HIV viral load, tuberculosis diagnosis, and other assays (though equipment sharing across programs is not common practice in the study setting). In Tanzania, two variations of the Xpert analyzer were available: Xpert II with the capacity to run two and Xpert IV four assays concurrently. In general, Xpert IV analyzers were placed at higher volume sites. In Tanzania, PoC analyzers were located in the pediatric outpatient clinic or health facility laboratory. In Mozambique, PoC analyzers were

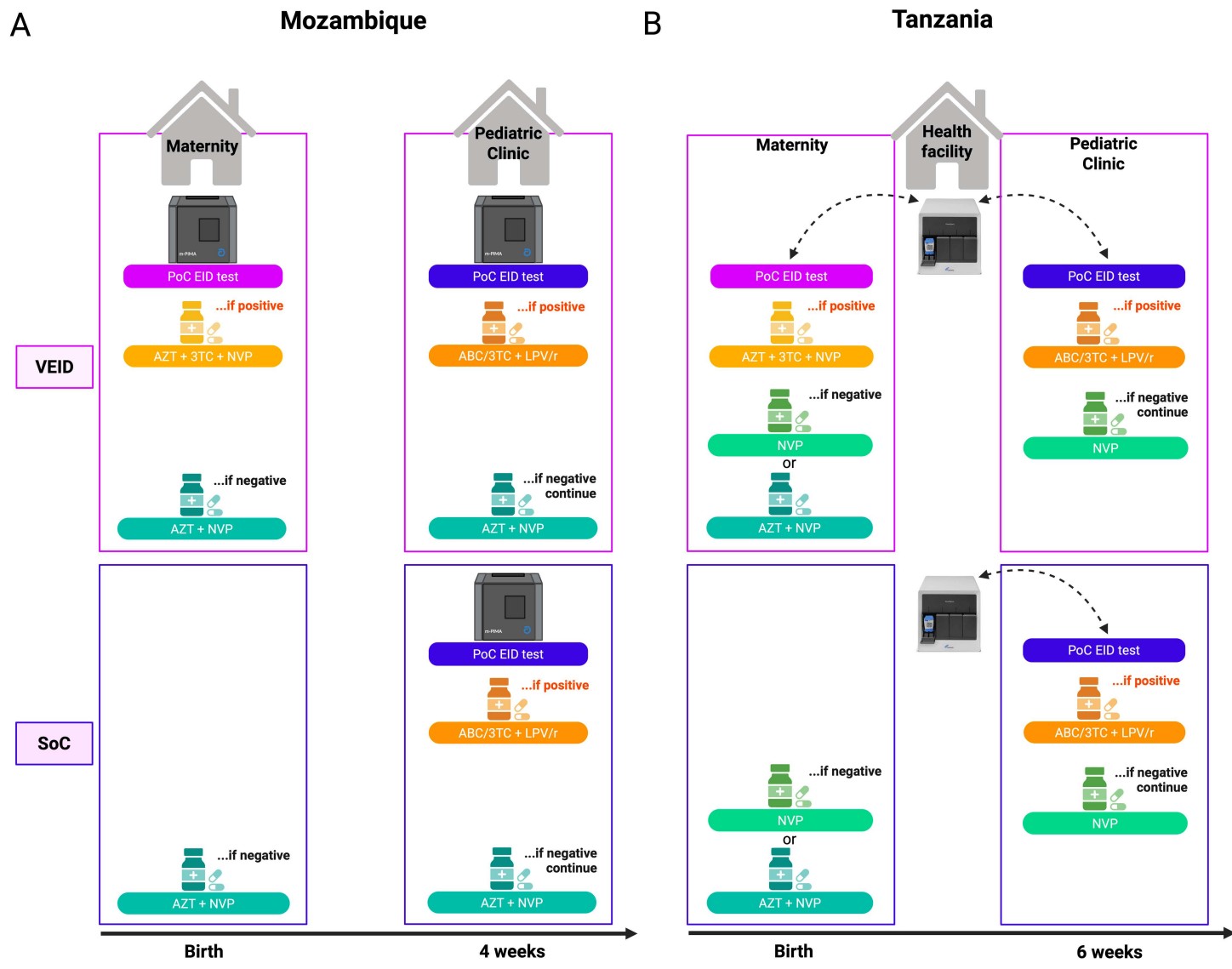

**Fig 1. Testing and treatment algorithms.** HIV-exposed infants received their first HIV test in the maternity ward (VEID) or the pediatric clinic (SoC), where follow-up visits were also performed. **(A)** In Mozambique, each clinic had one mPIMA analyzer. **(B)** In Tanzania, each health facility had one Xpert analyzer. Infants with HIV–positive test results were immediately initiated on ART. Infants with HIV–negative test results were initiated on post-natal pro-phylaxis according to country guidelines: AZT+NVP for all infants in Mozambique and high-risk infants in Tanzania and NVP only for low-risk infants in Tanzania.VEID, very early infant (HIV) diagnosis; SoC = standard of care; PoC EID, point-of-care early infant (HIV) diagnosis; AZT, zidovudine; 3TC, lami-vudine; NVP, nevirapine; ABC, abacavir. Created in BioRender. Hoelscher, *M.* (2025) https://BioRender.com/t4btr5b.

placed at the pediatric outpatient clinic, and an additional analyzer was placed at the maternity ward in VEID sites. Both analyzers have comparable diagnostic performance and run-time for EID [19,20]. Therefore, we expect that the choice of platform did not affect linkage to ART or other health outcomes.

## Costing approach

We conducted a micro-costing study from the health system perspective. We estimated implementation and operations costs as they occurred in the LIFE study, including fixed costs (equipment purchase, installation, and maintenance;

utilities; communication; training; and facility upgrades) and variable costs (consumables, labor, and ART) for VEID and SoC approaches. Direct client and societal costs were excluded from this analysis, as the study was designed to focus on direct budgetary and resource allocation impact on the healthcare provider. The time horizon for individual-level cost and effectiveness outcomes corresponded to the 12-week follow-up period, during which clinical pathways and resource use differed between arms. Beyond this period, costs and clinical management were assumed to be equivalent in both arms. Infants lost to follow-up after birth in the SoC arm incurred no intervention-related resource use and were therefore treated as structural zeros with respect to costs.

We first identified resources used for VEID from study documents, local HIV testing and care guidelines, and communication with implementing partners. To measure quantities of resources consumed, we used data from the LIFE study and usage data from the PoC analyzers. Consumables, labor, and ART were collected using a bottom-up approach. Time required for nurses to conduct PoC EID tests and perform HIV-related counselling was collected as anonymous self-reported average active work time once they had logged sufficient experience with PoC testing (>50 tests/individual). Fixed costs were allocated to EID testing according to observed testing volume; consumables, labor, and ART costs were allocated directly per test or per infant. Facility upgrades included the purchase of cabinets to store reagents and the installation of air conditioning units in some sites. Fixed costs were shared with routine services, including HIV viral load monitoring. Shared resources (e.g., analyzers, utilities, and communication) were allocated proportionally based on observed PoC EID testing volume relative to total analyzer use.

Prices for resources were collected from Ministry of Health budgets and expenditure reports, purchase tenders, manufacturer contracts, and government salary scales. Test cartridges were procured at a flat rate inclusive of shipping, customs clearance, distribution, and administration costs. Capital costs were discounted at 3% per year and amortized using equivalent annual cost—PoC platform costs over a 5-year life span according to manufacturer specifications and other capital costs over 10 years. Cost data collected for 2020 and 2021 were converted to nominal 2020 United States dollars (US$) using World Bank Global Economic Monitor annual mean exchange rates [21]. At the time of the study, these were 66.8 Mozambique Metical and 2304.4 Tanzanian Shillings per 1 US$ for 2020 and 64.4 Mozambique Metical per 1 US$ for 2021. No costs paid in Tanzanian Shillings were collected for 2021. Further costing details are provided in S1 Text.

## Outcomes

We assessed the proportion of infants with HIV initiating ART within 1 week of life, defined as trial-documented initiation of ART on or before day seven after birth, as the primary effectiveness outcome. We also assessed the proportion of infants exposed to HIV receiving an EID test within eight weeks of life as a secondary effectiveness outcome. EID testing was documented in study visit records and confirmed with PoC analyzer logs. Incremental costs, incremental effectiveness, and incremental cost-effectiveness ratios (ICERs) per additional infant initiating ART within 1 week and per additional infant receiving a PoC EID test within 8 weeks were estimated relative to the SoC. In the absence of robust data quantifying the incremental survival benefit associated with starting ART in the first week relative to starting at 4–8 weeks of life, we conducted a break-even analysis to derive the minimum life-years that would need to be gained per additional infant initiating ART within one week of life for our intervention to be considered cost-effective given standard cost-effectiveness thresholds.

## Statistical analysis

Frequentist methods were used for descriptive summaries and unadjusted comparisons between study arms. Descriptive cost per test and per infant were reported as testing volume-weighted mean and bootstrapped 95% CI per country and study arm to reflect typical costs across sites. To show the influence of testing volume on cost per test, we additionally presented unweighted medians by low-, medium-, and high-volume sites. Summary measures for effectiveness outcomes were reported as proportions for the probability of EID testing and ART initiation and median and range for age at first EID test and ART initiation.

Incremental costs and incremental effectiveness were estimated using multivariate Bayesian hierarchical models to account for skewed distribution of cost data and clustering at the health facility level, with results reported as posterior mean and 95% credible interval (CrI). Costs were modeled on the natural scale using a hurdle Gamma specification to accommodate observed zeros arising from infants who did not receive EID testing due to loss to follow-up, while effectiveness outcomes were modeled as Bernoulli processes. Models were fit separately for 1-week ART and 8-week PoC EID effectiveness outcomes with country- and site-level random effects to capture intra-cluster correlation and differences in costs, PoC testing platforms, prophylaxis protocols, and transmission rates. For the 1-week ART model, site-level random effects were not included due to sparse outcome variation within clusters, which precluded reliable estimation of additional variance components. The country was included as a fixed effect to account for the systematic differences in context. Informative priors were specified for study arm effects to reflect expected differences between VEID and SoC arms. Prior sensitivity analyses using weakly informative priors were conducted to assess the robustness of model estimates to prior specification. Model convergence was confirmed (all $R \approx 1$), and posterior predictive checks indicated good agreement between observed and replicated data. Full model specification, priors, and diagnostics are provided in S1 Text.

ICER point estimates were calculated as the ratio of mean incremental cost to mean incremental effect from the posterior distributions. To reflect uncertainty around these estimates, we used posterior draws of incremental cost and incremental effect jointly sampled from the Bayesian hierarchical models in probabilistic sensitivity analyses. Uncertainty was visualized in cost-effectiveness planes showing additional costs versus additional benefit for VEID over SoC. The probability of each PoC EID approach being cost-effective at a range of WTP thresholds is shown in cost-effectiveness acceptability curves. To contextualize results, we referenced empirically derived country-level cost-effectiveness thresholds from Pichon-Riviere and colleagues [22], which are based on health expenditure per capita and life expectancy growth and expressed as cost per life-year gained. These were $189 per life-year in Mozambique and $316 per life-year in Tanzania (2020 US$). For comparability with previous studies, we also considered GDP per capita as a willingness-to-pay (WTP) threshold ($462 in Mozambique and $1,117 in Tanzania per life-year gained) [21]. As ICERs in this analysis are expressed in cost per additional infant initiating ART within 1 week of life rather than per life-year gained, they cannot be directly compared to these thresholds. Rather than model long-term health outcomes directly, we conducted a break-even analysis to derive the minimum life-years that would need to be gained per additional infant initiating ART within one week of life for VEID to meet these thresholds, calculated as the posterior draw of incremental cost divided by the WTP threshold. Results are reported as median and 95% CrI.

Cost data was compiled in Microsoft Excel (Microsoft Corp.). Statistical analysis was performed using R version 4.2.3 (R Foundation for Statistical Computing, Vienna, Austria). Analyses followed a pre-specified health economic analysis plan. This study is reported as per the Consolidated Health Economic Evaluation Reporting Standards 2022 (CHEERS 2022) Statement [23] (S1 Checklist). Figures were created with R and BioRender.com.

## Sensitivity analyses

Deterministic sensitivity analyses were utilized to explore the impact of selected drivers of cost per test not explicitly parameterized in the Bayesian hierarchical models. We chose ranges reflective of plausible uncertainty and policy-relevant scenarios. Consumable prices were varied by ±50% to account for uncertainty in procurement and distribution costs, testing volume from −50% to 200% of observed levels to reflect variability across sites and potential scale-up, analyzer life span from 4 to 15 years for a conservative lower bound and assumptions that PoC platforms can last longer with maintenance, and discount rate from 0% to 6% in line with standard guidance [24]. This analysis was designed to decompose cost-input uncertainty and identify key drivers of unit costs across policy-relevant scenarios, and is distinct from the probabilistic sensitivity analysis, which addresses decision uncertainty.

We additionally varied the intrauterine transmission rate across a wide range of values within a probabilistic sensitivity analysis framework to assess how cost-effectiveness would vary across epidemiological contexts and to identify threshold

levels of transmission at which VEID would meet common cost-effectiveness benchmarks. Additional probabilistic uncertainty around ICERs was captured in the main analysis and visualized in cost-effectiveness planes and acceptability curves. Here, we focused on scenario-driven variation.

## Results

### Age of early infant HIV diagnostic testing and ART initiation

In VEID sites, 100% of infants exposed to HIV had a PoC EID test by the recommended 8 weeks of age (median 1 day, interquartile range [IQR]: 0–1 days), compared with 84.4% in SoC sites (median 4.7 weeks, IQR: 4.4–6.4) (Fig 2A and Fig AA in S1 Text). Proportions of infants receiving PoC EID testing by eight weeks in SoC sites differed substantially by country with 91% in Mozambique and 75% in Tanzania. Repeat testing due to errors or confirmatory testing was performed for 9.5% versus 3.7% in Mozambique and 9.9% versus 8.7% in Tanzania of infants for VEID and SoC approaches, respectively.

In VEID sites, 98.6% of infants diagnosed with HIV started ART compared to 89.3% in SoC sites, and among infants with an HIV–positive result available at birth (VEID sites), 89.5% started ART within the first week of life. The median age at ART initiation was 0.9 (range: 0–14) weeks in VEID sites and 4.7 (range: 4–16) weeks in SoC sites (Fig 2B and Fig AB in S1 Text). Reasons for not starting ART were death or loss to follow-up. Reasons for delayed treatment initiation at birth were not meeting the minimum weight requirement for neonatal antiretroviral dosing and a delay in confirmatory HIV testing.

### Costs of very early infant diagnosis versus standard-of-care

Cost per test was $39.81 (95% CI [$37.44, $43.70]) for the VEID approach and $41.97 (95% CI [$39.04, $46.58]) for the SoC approach using mPIMA in Mozambique. Using Xpert in Tanzania, these costs were $33.91 (95% CI [$30.85, $40.92]) for VEID and $39.90 (95% CI [$34.30, $49.75]) for SoC. Lower cost per test for the VEID approach reflects higher testing

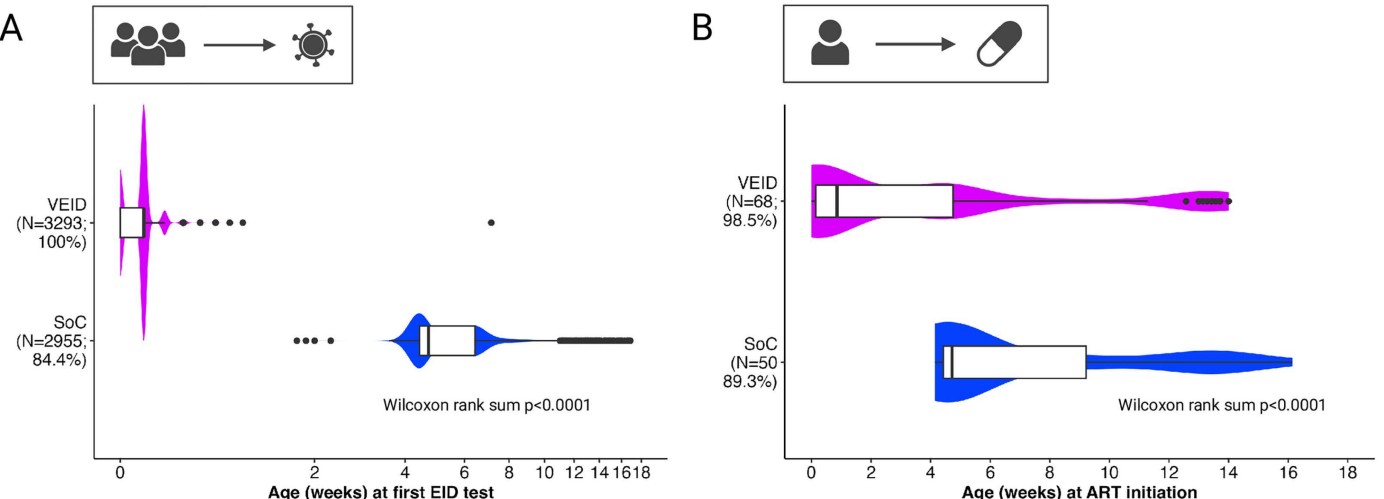

**Fig 2. Time from birth to PoC EID testing and treatment initiation. (A)** First EID test among all HIV-exposed infants; proportion receiving an EID test indicated in vertical axis labels. **(B)** ART initiation among infants diagnosed with HIV up to 16 weeks; proportion initiating ART at all indicated in vertical axis labels. Box plots show median (center line), 25% and 75% (box limits), and 5% and 95% (whiskers); violin plots show data distribution. VEID, very early infant (HIV) diagnosis; SoC = standard of care; PoC EID, point-of-care early infant (HIV) diagnosis; ART, antiretroviral treatment. Created in BioRender. Hoelscher, *M.* (2025) https://BioRender.com/u260ruo.

volumes, since infants were tested twice by design and fixed costs were allocated across more tests. Cost per infant exposed to HIV, which incorporates PoC EID testing and, if appropriate, earlier ART initiation was $90.13 (95% CI [$88.76, $91.47]) for VEID versus $39.70 (95% CI [$39.00, $40.42]) for SoC in Mozambique and $67.81 (95% CI [$66.26, $69.43]) for VEID versus $35.92 (95% CI [$34.67, $37.20]) for SoC in Tanzania. Cost inputs used to derive unit costs per test and per infant are listed in Table 1.

Consumables, primarily the test cartridge, accounted for 63% in Mozambique and 58% in Tanzania of the cost per test (Fig 3, Fig B and Table A in S1 Text). Equipment and overhead costs, by contrast, varied substantially by testing volume. We categorized health facilities into high (>12 tests per week), medium (5–12 tests per week) and low (<5 tests per week) volume for ease of comparison (S1 Text). On average in Mozambique, four (29%) sites ran over 15 tests weekly and two (14%) ran less than five tests weekly. In Tanzania, only one (7%) site ran over 15 tests weekly and six (43%) ran less than five tests weekly. Cost per test varied by 33% in Mozambique and 63% in Tanzania between high and low volume sites, with the greater variability in Tanzania driven by a larger number of low volume sites compared to Mozambique. Labor and overhead costs were small in comparison to consumable and equipment costs in both countries.

## Cost-effectiveness of very early infant HIV diagnosis

VEID was associated with higher costs and improved effectiveness outcomes in both countries (Fig 4, Tables B-C in S1 Text). In Mozambique, VEID increased the proportion of infants initiating ART within one week of life by 90.0 (95% CrI [67.5, 98.5]) percentage points at an incremental cost of $2,632 (95% CrI [$2,249, $3,062]) per infant with HIV. In Tanzania, the incremental cost was higher at $6,263 (95% CrI [$5,394, $7,243]), with a 59.9 (95% CrI [20.9, 89.5]) percentage point increase in the proportion initiating ART within one week. The corresponding ICER with respect to additional infants initiating ART within one week of life was $2,924 in Mozambique and $10,458 in Tanzania.

Based on the break-even analysis, VEID would need to achieve 15.24 (95% CrI [12.63, 21.37]) life-years gained per additional infant initiating ART within 1 week of life in Mozambique and 32.17 (95% CrI [21.15, 94.84]) in Tanzania to meet empirically derived cost-effectiveness thresholds [22] (Fig 5). At WTP thresholds based on 1x GDP per capita, the corresponding required life-years gained were 6.24 (95% CrI [5.17, 8.74]) in Mozambique and 9.10 (95% CrI [5.98, 26.83]) in Tanzania.

Implementation of VEID also increased uptake of EID. In Mozambique, VEID increased the proportion of infants receiving an EID test within eight weeks of life by 8.0 (95% CrI [5.4, 11.2]) percentage points at an incremental cost of $50.72 (95% CrI [$43.72, $58.65]) per infant exposed to HIV. In Tanzania, the corresponding increase was 20.9 (95% CrI [14.9, 28.0]) percentage points in the proportion of infants receiving an EID test within eight weeks of life at an incremental cost of $28.52 (95% CrI [$24.39, $33.19]) per infant exposed to HIV. The ICER with respect to additional infants receiving an EID test within eight weeks of life was $635.47 and $136.19 in Mozambique and Tanzania, respectively.

## Sensitivity analysis

Cost per test was most sensitive to relative changes in the price of consumables, followed by testing volume and PoC analyzer life span (Fig 6 and Fig C in S1 Text). We observed moderate variability in baseline cost across health facilities resulting from different testing volumes and 5% probability of zero cost across all sites (i.e., infants not receiving testing mostly in SoC sites due to loss to follow-up after birth). Site-level variation was modest with an intra-cluster correlation of 0.033 (95% CrI [0.018, 0.059]) for cost per infant and 0.140 (95% CrI [0.066, 0.258]) for PoC EID within eight weeks. Sites with higher baseline costs tended to experience smaller increases in cost per additional test as testing volumes increased (posterior correlation ρ = −0.25), consistent with economies of scale. Due to low number of infants diagnosed with HIV in Tanzania, uncertainty in effectiveness outcomes was high; combined with uncertainty in costs, this resulted in substantial uncertainty in the ICER with respect to ART initiation in the first week of life in Tanzania.

**Table 1. Costs of inputs. Costs expressed in 2020 US$ by category for each country/PoC testing platform.**

| | Mozambique mPIMA | Tanzania Xpert | Source |
|---|---|---|---|
| Fixed and capital costs | | | |
| Equipment | | | |
| PoC platform purchase price | $15,000 | Xpert IV $17,500 Xpert II $12,280 | CHAI/ Cepheid |
| PoC platform maintenance (per year; years 2–5) | $2,500 | Xpert IV $1,000 Xpert II $500 | |
| Undiscounted PoC platform cost (5 years) | $25,000 | Xpert IV $21,500 Xpert II $14,280 | |
| Discounted PoC platform cost (5 years) | $23,585 | Xpert IV $20,599 Xpert II $13,727 | |
| Facility upgrades | | | |
| Cabinets for equipment and consumable storage per site | $224 | $1,996[b] | CHAI |
| Air conditioning units average per site[a] | $265 | | |
| Other upgrades per site | $143 | | |
| Undiscounted cost per site (10 years) | $632 | $1,996 | |
| Discounted cost per site (10 years) | $614 | $1,938 | |
| Training | | | |
| Undiscounted cost per site (5 years) | $213 | $252 | CHAI |
| Discounted cost per site (5 years) | $207 | $245 | |
| Overhead | | | |
| Electricity share per site (1 year) | $27.95 | $40.47 | MoH |
| Communication share per site (1 year) | $394 | $260 | |
| Total undiscounted annual fixed costs | **$6,001** | **Xpert IV $5,284 Xpert II $3,708** | |
| Total discounted annual fixed costs | **$5,689** | **Xpert IV $5,079 Xpert II $3,578** | |
| Variable costs | | | |
| Consumables | | | |
| Test cartridge[c] | $25.00 | $21.87 | CHAI |
| Neonatal sample collection | $1.20 | $1.50 | |
| Labor | | | |
| Personnel cost per test | $0.77 | $1.33 | Gov. salary scales and self-reported time use |
| Antiretrovirals (per week) | | | |
| 3TC syrup | $2.10 | .. | CHAI/WHO |
| AZT/3TC disp. tablets | .. | $1.85 | |

[a]Not all sites had air conditioning units installed; [b] Available only as an aggregate figure; [c] Inclusive of shipping, custom clearance, distribution, and administrative costs; PoC = point-of-care; CHAI = Clinton Health Access Initiative; MoH = Ministry of Health; Gov. = Government; AZT = Zidovudine; 3TC = Lamivudine; NVP = Nevirapine; disp. = dispersible; WHO = World Health Organization.

Intrauterine transmission rates observed in the LIFE study were 1.01% (95% CrI [0.65, 1.51]) in SoC sites and 1.48% (95% CrI [1.01, 2.06]) in VEID sites in Mozambique and 0.43% (95% CrI [0.16, 0.85]) in SoC sites and 0.50% (95% CrI [0.19, 0.94]) in VEID sites in Tanzania. The ICER with respect to additional infants initiating ART within one week of life was sensitive to intrauterine transmission rate, ranging from $2,394 to $4,009 in Mozambique and from $5,993 to $27,424 in Tanzania between the 5th and 95th percentiles of estimated intrauterine transmission. In Mozambique, intrauterine transmission would need to exceed 34% to meet the cost-effectiveness threshold from [22] or 14% to meet

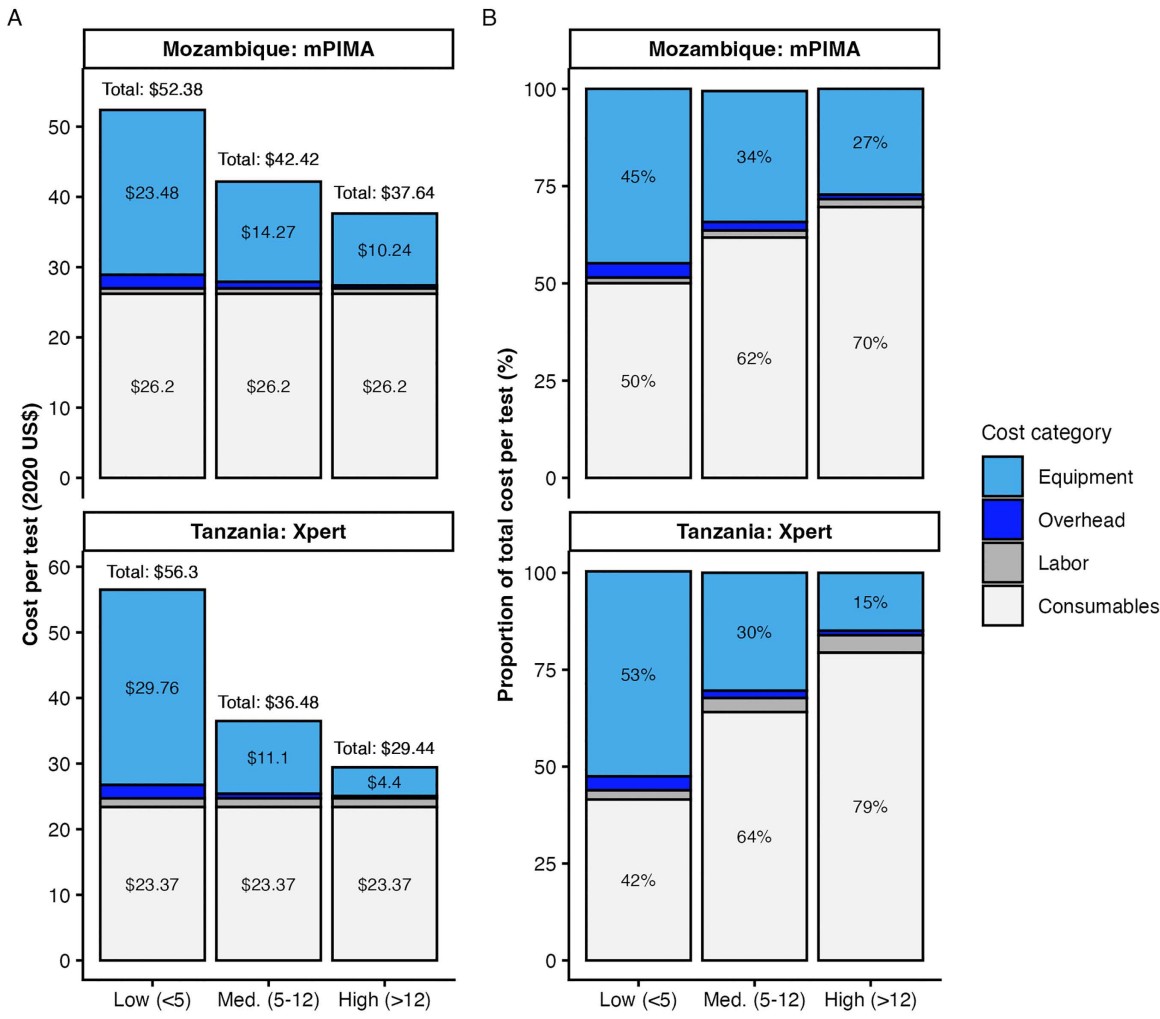

**Fig 3. PoC EID test cost components. (A)** Cost estimates expressed in 2020 US$. **(B)** Proportion of total cost per test. Sites are categorized as high (>12 tests per week), medium (5–12 tests per week), and low (<5 tests per week) testing volume. Equipment includes amortized costs of initial purchase, installation, and yearly maintenance of PoC analyzers. Overhead includes apportioned costs of electricity, communications, and facility upgrades. Overhead and labor costs were < $3 per test (<6% of cost per test) across all sites.

the GDP-based threshold (Fig 7). In Tanzania, the corresponding intrauterine transmission rates were 20% and 5.8%, respectively, indicating that VEID becomes cost-effective only above specific thresholds of early vertical transmission risk. These thresholds are based on cost per additional infant initiating ART within 1 week of life and do not directly incorporate downstream life-years gained, which are explored separately above.

## Discussion

Focusing on primary healthcare settings in Mozambique and Tanzania, this trial-based economic evaluation estimated the cost and cost-effectiveness of offering PoC HIV diagnostic services at birth and 4–8 weeks of age compared to the SoC at 4–8 weeks only. We evaluated intermediate outcomes, ART initiation within 1 week of life and EID uptake within 8 weeks of life, which are important for reducing early mortality and morbidity. In the context of uncertain global funding for HIV programs, particularly recent funding freezes and proposed cuts to the President's Emergency Plan for AIDS Relief

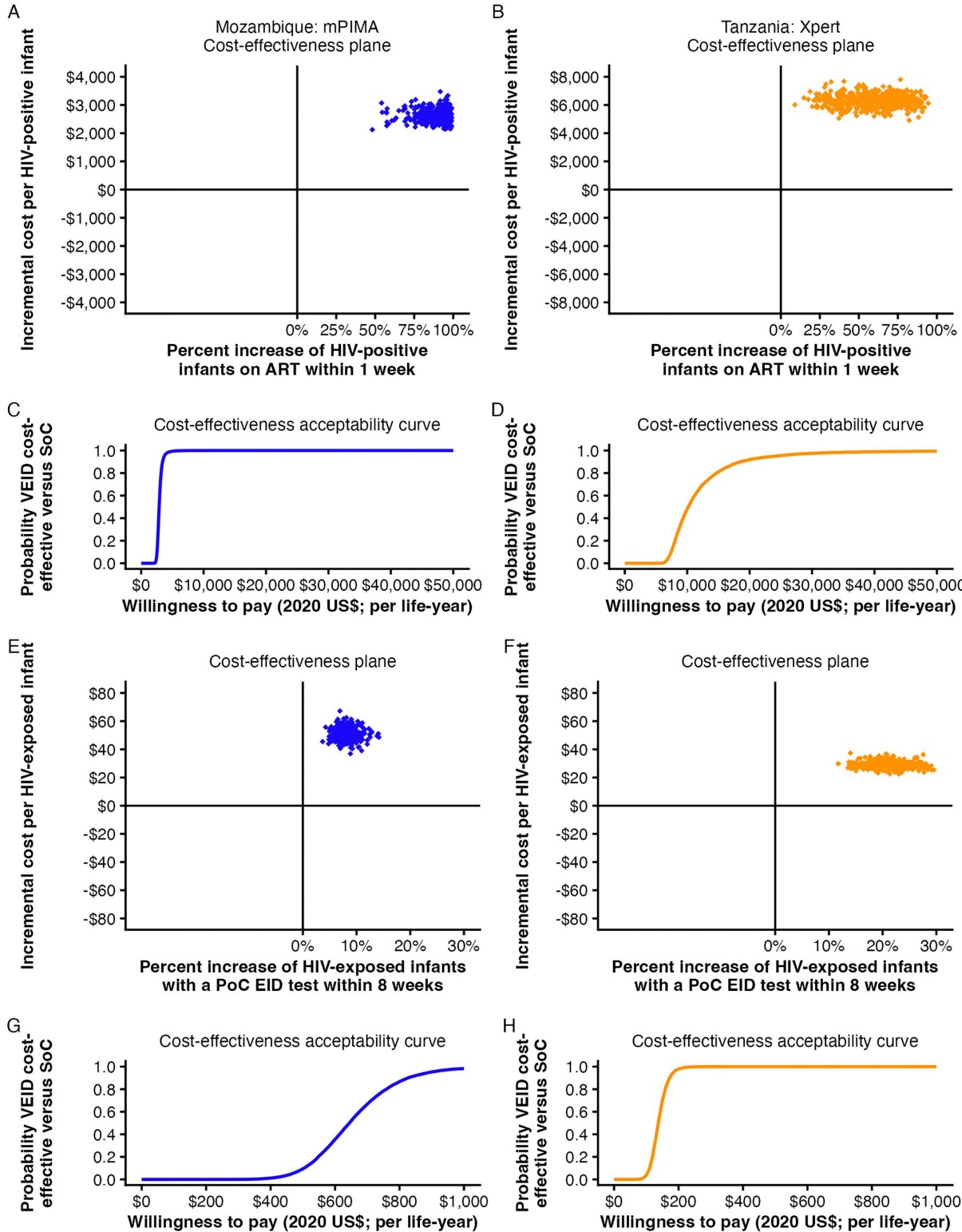

**Fig 4. Cost-effectiveness of VEID with respect to PoC EID testing and early ART.** Cost-effectiveness planes presenting the incremental costs and (**A** and **B**) early ART benefit (% of eligible infants with HIV initiated on ART within 1 week of life; eligible infants include those meeting clinical criteria for

treatment, e.g., weight) or 8-week EID benefit (%) for VEID **(E** and **F)**. Points represent estimates of costs and health benefit for VEID relative to SoC at the origin. A random sample of 500 points is plotted. All estimates indicate that VEID is more expensive and results in increased health benefit (upper right quadrant). **(C and D; G and H)** Cost-effectiveness acceptability curves for each outcome in A-B and E-F, respectively, showing cumulative probabilities of each testing strategy being cost-effective at a particular willingness to pay value. Costs are presented as 2020 US$. PoC EID, point-of-care early infant (HIV) diagnosis; ART, antiretroviral treatment; SoC = standard of care; VEID, very early infant (HIV) diagnosis.

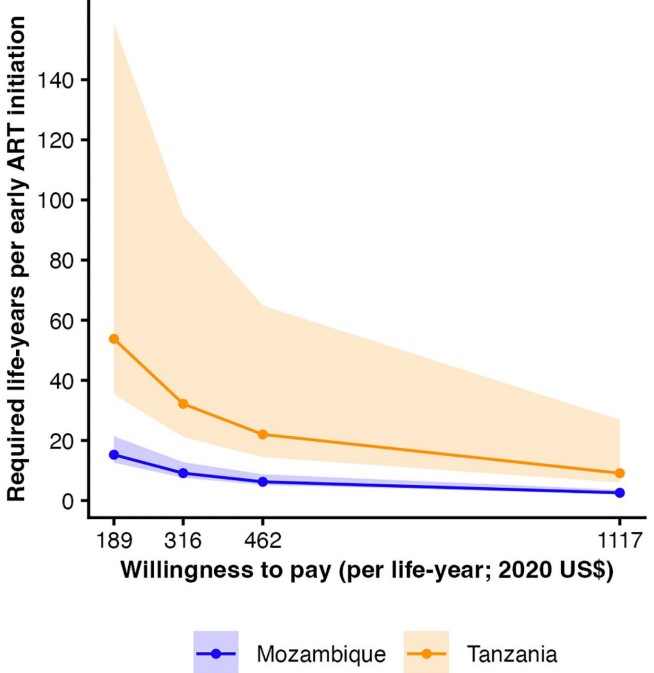

**Fig 5. Required life-years per early ART initiation for cost-effectiveness.** Shaded ribbons for Mozambique and Tanzania indicate 95% credible interval (CrI) across posterior draws from the Bayesian hierarchical model for 1-week ART initiation. Empirically derived cost-effectiveness thresholds, based on health expenditure and life expectancy growth [21] and expressed in 2020 US$, were $189 in Mozambique and $316 in Tanzania. 2020 gross domestic product per capita-based thresholds were $462 in Mozambique and $1,117 in Tanzania. Higher required life-years reflect scenarios in which greater health gains per infant would be needed to approach the indicated thresholds.

(PEPFAR), the landscape of HIV epidemiology and vertical transmission may change substantially [25,26]. Amid this uncertainty, generating evidence on the cost and cost-effectiveness of VEID is crucial for guiding investment and design decisions in EID programs, especially as transmission rates and resource availability evolve.

VEID at birth increased the proportions of infants tested and initiated on ART and reduced the age at ART initiation. Previous studies also show PoC testing of newborns in similar settings results in earlier ART initiation [27–29], including a multi-country observational study which reported that 92.3% of infants with HIV diagnosed at the PoC initiated ART within 60 days of sample collection [9]. In our study, the majority of infants delivered at VEID sites received a PoC HIV test in the first 24 hours of life and initiated ART within the first week of life. Initiating ART shortly after birth may suppress viral replication and inhibit the establishment of long-lasting viral reservoirs, which could slow disease progression and reduce mortality and morbidity [12,13], as also observed in the LIFE study [15].

From a cost perspective, our findings were consistent with other PoC EID studies using comparable testing platforms and including equipment purchase [30]. Cost per test was driven by reagent costs, apart from at sites with very low testing volumes. These results echo advocacy initiatives, such as the *Time for $5* campaign, which emphasize the importance of

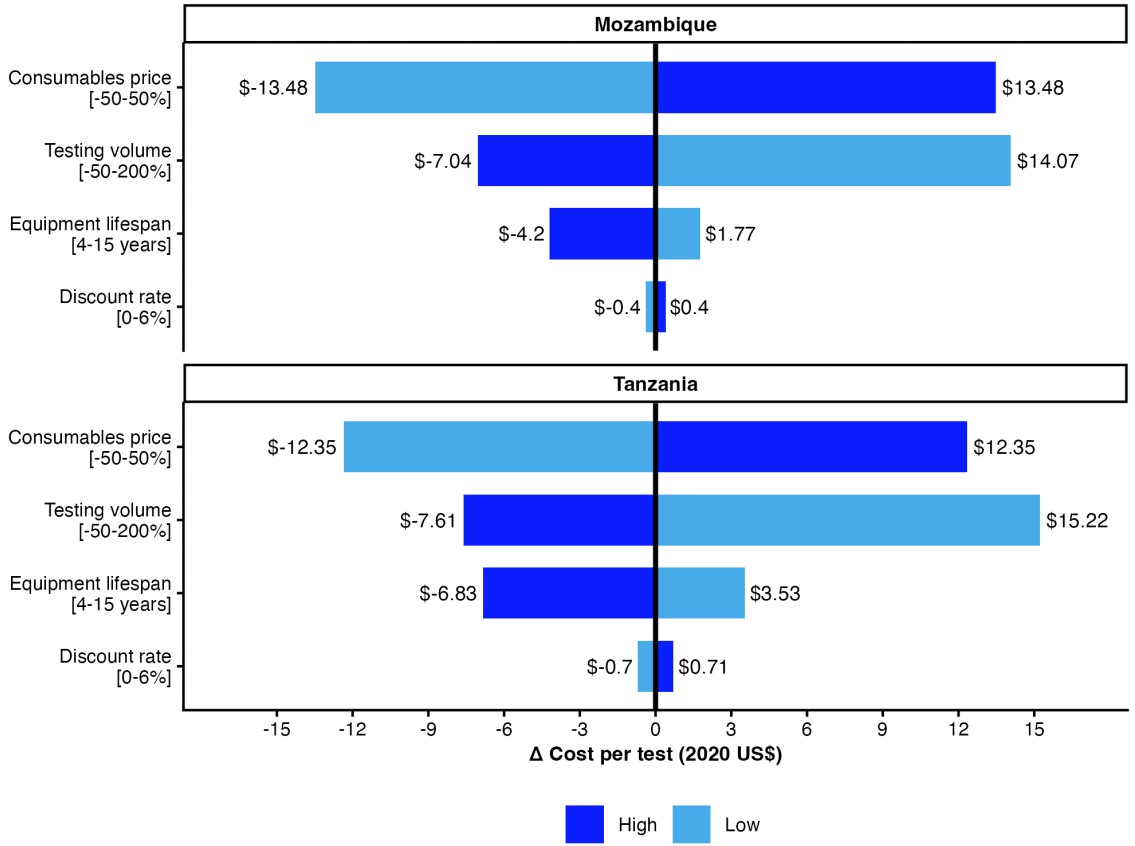

**Fig 6. Deterministic one-way sensitivity analysis showing influence of key assumptions on cost per PoC EID test.** Costs expressed in 2020 US$. Per test cost difference is shown on the x-axis and parameters with ranges are shown on the y-axis. High and low colors indicate the direction in which the parameter varies. This analysis addresses cost-input uncertainty rather than decision uncertainty; see Figure 4 for probabilistic sensitivity analysis of incremental cost-effectiveness.

lowering reagent prices for sustainable access in low- and middle-income countries [31]. We included equipment, over-head, labor, and consumable costs in our calculations, which explains higher cost per test compared to studies omitting equipment procurement and set-up [32,33]. Without initial equipment investment (i.e., in scenarios where existing PoC testing infrastructure allows for repurposing of analyzers for PoC EID or integration with other programs), cost per test can be reduced by up to 43% for mPIMA in Mozambique and 47% for Xpert in Tanzania. However, as manufacturers specify relatively short analyzer lifespans, estimating costs with initial equipment investment for EID remains important for comprehensive budget planning. Further, cross-utilization of PoC infrastructure across programs (e.g., HIV viral load monitoring, tuberculosis diagnosis) may be a viable approach to increase testing volume and decrease costs, especially at low-volume sites. At 70% utilization, cost per test was reduced by up to 4% for mPIMA in Mozambique and 27% for Xpert in Tanzania, highlighting an advantage of the Xpert platform's broad multiplex diagnostic capability. In both countries, leveraging existing PoC infrastructure and increasing PoC analyzer utilization can roughly halve costs and yield substantial efficiency gains.

Previous modelling studies have demonstrated that PoC compared to laboratory-based EID is cost-effective, increases the proportion of infants rapidly initiated on ART, and improves life expectancy across a range of programmatic and economic conditions [32,34]. While we did not assess cost-effectiveness of PoC versus laboratory-based EID, our estimated

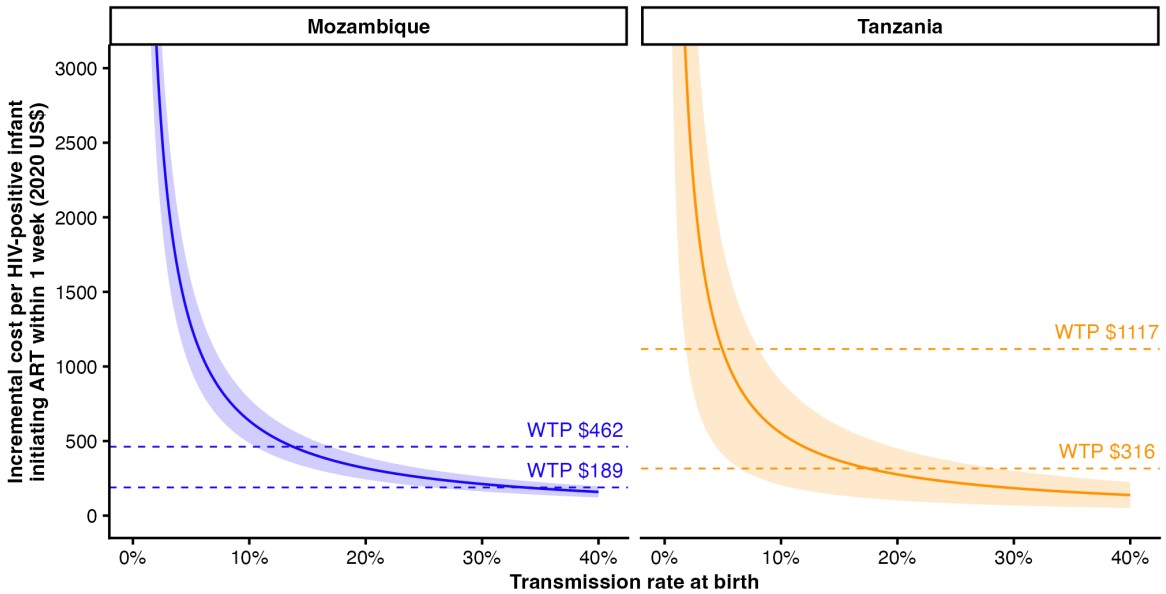

**Fig 7. Sensitivity analysis of the incremental cost effectiveness ratio (ICER) per infant with HIV on ART within 1 week across a range of intrauterine transmission rates, based on posterior estimates from the Bayesian hierarchical model.** Lines show the posterior mean ICER, with shaded areas representing the 95% credible intervals. Dashed horizontal lines indicate reference willingness to pay (WTP) thresholds: empirically derived cost-effectiveness thresholds based on health expenditure and life expectancy growth from [21], expressed in 2020 US$ ($189 for Mozambique and $316 for Tanzania) and gross domestic product per capita in 2020 ($462 for Mozambique and $1,117 for Tanzania).

costs of PoC EID were comparable to costs of laboratory-based EID tests reported in other studies [30]. Instead, we compared PoC birth plus 4–8 week testing with PoC 4–8 week testing alone, consistent with WHO guidelines [3]. This likely underestimated the added impact of VEID, as SoC also benefited from shorter PoC turnaround times compared with central laboratory-based testing in our study. A model-based analysis of birth plus 6-week testing in South Africa improved survival and was deemed cost-effective as long as uptake was high [16]. Our study adds real-world experience related to test frequency and timing from two countries with different intrauterine transmission rates using different PoC testing platforms.

Although early ART initiation is an intermediate outcome, it is an important precursor to survival benefit [14,35]. The survival benefits required for VEID to be considered cost-effective based on the break-even analysis, 15–32 life-years gained per additional infant initiating ART within 1 week using cost-effectiveness thresholds derived from Pichon Riviere and colleagues [22] or 6–9 life-years gained using GDP-based thresholds, are substantially higher than those demonstrated to date. In the CHER trial [14], which remains the primary benchmark, early versus deferred ART was associated with roughly 0.8 life-years gained over 5 years, based on reported deaths and follow-up time per arm. However, CHER compared 6-week to median 7-month ART initiation, while our study compared birth to 4–8-week ART initiation. Importantly, HIV-related infant mortality peaks around 2–3 months [5], suggesting that ART at birth could be underestimated. While early mortality reductions in the LIFE study were not sustained through 18 months, most infants received LPV/r and viral suppression was low and associated with poor clinical outcomes overall [15]. Differences in treatment regimens, adherence patterns, background mortality, and health system context across settings and time introduce structural uncertainty that may influence the magnitude of survival benefits achievable with ART initiation in the first week of life. With dolutegravir-containing ART now available for infants and expected to address common barriers to adherence due to once daily dosing and improved palatability, greater longer-term health benefits may be achievable than observed in

earlier cohorts. Differences in comparator, follow-up, and treatment context caution against directly extrapolating CHER outcomes to our setting and highlight that long-term benefits of VEID are still uncertain. The life-years estimates above are a derived benchmark intended to contextualize the magnitude of survival benefit required for cost-effectiveness. The cost-effectiveness thresholds used are intended to approximate health system opportunity costs and should be considered alongside local budget constraints, feasibility, and equity priorities when informing policy decisions.

Beyond survival, transmission dynamics were central to the cost-effectiveness of VEID. Mozambique and Tanzania observed average intrauterine transmission rates of 1.3% and 0.5%, respectively, and the incremental cost per additional infant initiating ART within one week of life was sensitive to this. Our analysis indicates that intrauterine transmission would need to exceed 30% in Mozambique and 20% in Tanzania using an empirically derived cost-effectiveness threshold, or 14% and 5.8% using a GDP-based threshold, for universal VEID to be cost-effective. Settings with low intrauterine transmission rates may alternatively consider targeted VEID based on maternal risk factors (e.g., lack of ART, high viral load) [36], combined with efforts to maintain PoC analyzer utilization, as a more feasible path to affordability in low- and middle-income countries. In settings with higher intrauterine transmission or where survival benefits of VEID exceed current evidence, universal VEID may be more justifiable. Where and to whom VEID is offered may also have important equity implications. The benefits of VEID are likely concentrated among infants at higher risk of vertical transmission and in health facilities with sufficient capacity to reliably deliver timely testing and treatment. In settings with lower testing volumes or weaker follow-up systems, costs and benefits may differ, underscoring the importance of pairing VEID implementation with health system strengthening so that it reduces disparities in infant HIV testing and linkage to care.

Retention in care, adherence to treatment, and ensuring follow-up HIV testing for infants while they remain at risk for acquiring HIV, which depend on strong health systems and support services, are other important considerations when evaluating the cost-effectiveness of PoC EID approaches. Very early initiation of ART can only be translated into longer-term health benefits if infants continue receiving effective treatment. Thus, supportive interventions addressing other challenges such as inadequate family support, lack of disclosure, and postpartum maternal depression are also needed [37]. Additionally, all infants should receive follow-up testing to identify those with undetectable virus at birth or acquiring HIV during the breastfeeding period [3,16], especially considering potential disruptions to maternal ART access due to funding cuts. We assumed that infants lost to follow-up did not incur further program-related costs. If unobserved services were received elsewhere, this assumption could bias costs downward. However, loss to follow-up was low (<5%) in our study. While full population coverage and universal follow-up testing would slightly increase program costs, our estimates reflect what a well-run program could realistically achieve.

A strength of this study is that it is primarily informed by a trial combining PoC and birth EID designed to mirror routine care and align with country-specific HIV program priorities. We provided contextualized information about resources required to scale-up EID programs, about which little was known [16]. The limitations include that uncertainty estimates for many cost inputs were lacking. LIFE study recruitment spanned a time period when prices and health-seeking behavior may have been influenced by the COVID-19 pandemic. We mitigated price fluctuations by relying mostly on pre-pandemic costs and did not observe a notable decline in 4–8-week visit attendance during lockdown periods. Costs related to confirmatory central laboratory testing and broader shared infrastructure were not included, which may underestimate total programmatic costs, especially at low-volume sites. Our probabilistic sensitivity analysis propagates uncertainty arising from observed variation in costs and effectiveness outcomes captured by the Bayesian hierarchical models. We did not additionally model uncertainty in individual cost inputs, as the analysis was based on observed resource use and expenditures. In addition, scarce long-term health data for infants with HIV—particularly survival times or quality-of-life measures—make assessing the efficiency of interventions challenging. Given the absence of a significant difference in longer-term health outcomes in the LIFE study [15], we restricted our analysis to a 12-week time horizon and assessed cost-effectiveness using intermediate outcomes relevant to EID programs as key indicators of health benefit. We referenced empirically derived cost-effectiveness thresholds as well as GDP per capita [21,22,38], however, these thresholds,

expressed in terms of life-years gained rather than additional early ART initiations, should be used to contextualize results and not to directly interpret ICERs. Our break-even analysis provided an informative benchmark of the survival benefits required for cost-effectiveness, but these estimates were derived from posterior draws and depend on assumptions about WTP. This approach excludes explicit examination of cost-utility in terms of quality- or disability-adjusted life years. Dynamic, longer-term modeling frameworks could more explicitly account for longitudinal clinical transitions and age-dependent mortality, though such approaches would require substantial extrapolation beyond the observed data. Finally, direct client and broader societal costs (e.g., schooling gains) were beyond the scope of this work.

From a healthcare system perspective, universally offering PoC EID at birth to infants exposed to HIV was more expensive and resulted in more frequent and earlier ART initiation. Early ART initiation could help reduce persistently high HIV-related mortality and morbidity among infants. Our analysis suggests that VEID would need to generate substantial long-term health benefits to meet standard cost-effectiveness thresholds at current costs. The limited available evidence linking early ART initiation to longer-term health benefits suggests that these benefits may be smaller than required for VEID to be cost-effective when offered universally. The cost-effectiveness of VEID increases if ART initiation at birth translates to long-term survival benefits or if VEID can be effectively targeted to infants at high risk of HIV acquisition. In settings with high vertical transmission rates or where weak HIV programs lead to significant gaps in infant retention in care, EID programs could consider adding PoC birth testing with careful consideration of PoC infrastructure utilization to improve affordability. In settings with low vertical transmission rates, the additional benefit of universal birth testing is limited and targeted testing of high-risk infants at birth may be a more practical approach.

## Supporting information

**S1 Text. Supplementary material. Fig A: Violin plots of per site time from birth to PoC HIV EID testing and treatment initiation.** (A) First EID test among all HIV-exposed infants; (B) ART initiation among infants diagnosed with HIV up to 16 weeks of age. Only sites with infants with HIV are shown. Black triangle markers represent individual data. VEID, very early infant (HIV) diagnosis; SoC = standard of care; EID, early infant (HIV) diagnosis; ART, antiretroviral treatment. Created in BioRender. Hoelscher, M. (2025) https://BioRender.com/u260ruo. **Fig B: Per site point-of-care early infant (HIV) diagnosis test cost components.** (A) Cost estimates expressed in 2020 US$. (B) Proportion of total cost per test. **Fig C: Per site deterministic one-way sensitivity analysis showing influence of key assumptions on cost per test.** Per test cost difference is shown on the x-axis and parameters with ranges evaluated are shown on the y-axis. High and low colors indicate the direction in which the parameter varies. **Table A: Point-of-care early infant (HIV) diagnosis test cost components.** Cost estimates are expressed in 2020 US$. Sites are categorized as high (>12 tests per week), medium (5–12 tests per week), and low (<5 tests per week) testing volume. Equipment includes amortized costs of initial purchase, installation, and yearly maintenance of PoC analyzers. Overhead includes apportioned costs of electricity, communications, and facility upgrades. **Table B: Modeled incremental cost and effect estimates for 1-week ART uptake per infant diagnosed with HIV.** Estimated means and 95% CrI shown. SoC = standard of care; VEID, very early infant (HIV) diagnosis; CrI = credible interval. **Table C: Modeled incremental cost and effect estimates for 8-week PoC EID uptake per infant exposed to HIV.** Estimated means and 95% CrI shown. SoC = standard of care; VEID, very early infant (HIV) diagnosis; CrI = credible interval. **Fig D: Histogram of median PoC tests per ISO week** with divisions for low-, medium-, and high-volume sites shown by dashed lines. PoC = point-of-care; ISO, International Organization for Standardization. **Fig E: Model parameter traces** for A) 1-week ART uptake model and B) 8-week PoC EID uptake model. Burn-in phase not included. Parameters defined in 1. ART, antiretroviral treatment; PoC EID, point-of-care early infant (HIV) diagnosis. **Fig F: Posterior predictive plots** for A) Cost, B) 1-week ART uptake and C) 8-week PoC EID uptake. y refers to the observed data, yrep to the simulated data from the posterior predictions. ART, antiretroviral treatment; PoC EID, point-of-care early infant (HIV) diagnosis. **Fig G: Posterior distributions and scatter plots for 1-week ART uptake model.** ART, antiretroviral treatment. **Fig H Posterior distributions and scatter plots for 8-week PoC EID uptake**

**model.** PoC EID, point-of-care early infant (HIV) diagnosis. **Fig I: Distributions of intrauterine transmission probabilities** (defined as probability of positive HIV test result at birth) generated from 10,000 Bayesian bootstrap samples per country and study arm. SoC = standard of care; VEID, very early infant (HIV) diagnosis. **Table D: Summary of model parameters and posterior estimates. Table E: Posterior estimates for primary informative directional and sensitivity weak neutral prior specifications.** Parameter definitions can be found in Table D.
(PDF)

**S1 Checklist. CHEERS 2022 Checklist.** Husereau D, Drummond M, Augustovski F, de Bekker-Grob E, Briggs AH, Carswell C, Caulley L, Chaiyakunapruk N, Greenberg D, Loder E, Mauskopf J, Mullins CD, Petrou S, Pwu RF, Staniszewska S; CHEERS 2022 ISPOR Good Research Practices Task Force. Consolidated Health Economic Evaluation Reporting Standards 2022 (CHEERS 2022) statement: updated reporting guidance for health economic evaluations. BMJ. 2022 Jan 11;376:e067975. doi: https://doi.org/10.1136/bmj-2021-067975. PMID: 3550177145; PMCID: PMC8749494. This checklist is licensed under the Creative Commons Attribution 4.0 International (CC BY 4.0).
(DOCX)

## Acknowledgments

The authors gratefully acknowledge the infants and families who participated in the study and the dedicated healthcare personnel at the study sites. They also thank Martina Penazzato and Lara Vojnov from the World Health Organization, Landon Myer and Lynn Horn from the University of Cape Town, and Karim Manji from the Muhimbili University of Health and Allied Sciences in Dar es Salaam for external expert advice. The study was conducted at health facilities within the Mozambican and Tanzanian National HIV programs. The Clinton Health Access Initiative (CHAI) provided infrastructural support at the study sites and contingency supplies for pediatric antiretroviral drugs. We thank Lise Ellyin and Helder Mendes from CHAI Mozambique, and Patricia Mbago and Esther Mtumbuka from CHAI Tanzania for their support.

The authors also acknowledge the LIFE Study Consortium: Araújo Patricio, Dadirai Mutsaka, Lara Samuel, Sergey Bocharnikov, Timothy Bollinger, Wilson Simbine, Abhishek Bakuli, Cornelia Lueer, Elmar Saathoff, Fidelina Zekoll, Mariana Mueller, Friedrich Rieß, Otto Geisenberger, Nuno Taveira, Rute Marcelino, Absalao Zumba, Daniel Machavae, Adolfo Vubil, Ana Duajá, Jacinto Adolfo Ndarissone, Joao Manuel, Maria Maviga, Nalia Ismael, Jorge Morais, Nedio Mabunda, Adolfo Vubil, Fatima Mecupa, Amina de Sousa, Abisai Kisinda, Chacha Mangu, Doreen Pamba, Festina Paschal, Hellen Mahiga, Janeth Stephen, Lilian Njovu, Margareth Haule, Oliver Lyoba, Theodora Mbunda, Nhamo Chiwerengo, and Willyhelmina Olomi.

## Author contributions

**Conceptualization:** Kira Elsbernd, Issa Sabi, Ilesh V. Jani, Chishamiso Mudenyanga, Nyanda Elias Ntinyinya, Michael Hoelscher, Till Baernighausen, Arne Kroidl, Stefan Kohler.

**Data curation:** Kira Elsbernd, Chishamiso Mudenyanga, Siriel Boniface, Arlete Mahumane, Joaquim Lequechane, Falume Chale, Bindiya Meggi, Kassia Pereira, Raphael Edom.

**Formal analysis:** Kira Elsbernd.

**Funding acquisition:** Kira Elsbernd, Issa Sabi, Ilesh V. Jani, Nyanda Elias Ntinyinya, Michael Hoelscher, Till Baernighausen, Arne Kroidl, Stefan Kohler.

**Investigation:** Siriel Boniface, Arlete Mahumane, Joaquim Lequechane, Falume Chale, Bindiya Meggi, Kassia Pereira, Raphael Edom, Anange F. Lwilla.

**Methodology:** Kira Elsbernd, Issa Sabi, Ilesh V. Jani, Chishamiso Mudenyanga, Anange F. Lwilla, Till Baernighausen, Stefan Kohler.

**Project administration:** Bindiya Meggi, Stefan Kohler.

**Resources:** Ilesh V. Jani, Nyanda Elias Ntinyinya, Michael Hoelscher.

**Supervision:** Issa Sabi, Ilesh V. Jani, W. Chris Buck, Nyanda Elias Ntinyinya, Arne Kroidl, Stefan Kohler.

**Validation:** Kira Elsbernd, Issa Sabi, Ilesh V. Jani, Chishamiso Mudenyanga, Bindiya Meggi, Anange F. Lwilla, W. Chris Buck, Arne Kroidl, Stefan Kohler.

**Visualization:** Kira Elsbernd.

**Writing – original draft:** Kira Elsbernd.

**Writing – review & editing:** Kira Elsbernd, Issa Sabi, Ilesh V. Jani, Chishamiso Mudenyanga, W. Chris Buck, Arne Kroidl, Stefan Kohler.

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
