## [Editor Report · Decision Letter 0]

28 Feb 2025

Dear Dr Elsbernd,

Thank you for submitting your manuscript entitled "Comparative cost and cost-effectiveness of point-of-care early infant diagnosis at birth: Findings from a pragmatic cluster-randomized trial in Mozambique and Tanzania" for consideration by PLOS Medicine.

Your manuscript has now been evaluated by the PLOS Medicine editorial staff as well as by an academic editor with relevant expertise and I am writing to let you know that we would like to send your submission out for external peer review.

Please include the study protocol as supplementary information, as well as a completed CHEERS checklist. Please note that the original report in reference 15 of the trial results indicates that testing occurred at 4-8 weeks and not 4-6 weeks. Please resolve the discrepancy.

Please discuss the study limitations and include a discussion of the results in reference 15 regarding the lack of reduction of clinical outcomes at 18 mo associated with very early testing and acknowledge the other challenges (adherence) that must be addressed. Please also provide some discussion surrounding the high cost of very early testing to LMICs, and a discussion of how this study adds to the existing literature.

Please re-submit your manuscript within two working days, i.e. by Mar 04 2025 11:59PM.

Kind regards,

Alison Farrell, Ph.D.

Senior Editor

PLOS Medicine

---

## [Decision Letter · Decision Letter 1]

7 Aug 2025

Dear Dr Elsbernd,

Many thanks for submitting your manuscript "Comparative cost and cost-effectiveness of point-of-care early infant HIV diagnosis at birth: Findings from a pragmatic cluster-randomized trial in Mozambique and Tanzania" (PMEDICINE-D-25-00626R1) to PLOS Medicine. I apologize for the delay in conveying to you our decision, which has been due to difficulty in acquiring sufficient breadth of reviewer expertise. The paper has now been reviewed by subject experts and a statistician; their comments are included below and can also be accessed here: ********

As you will see, the reviewers find the work interesting, but have indicated that numerous additional analyses, including sensitivity analyses, would strengthen the study, and its rationale and conclusions need to be better justified and presented, respectively. I have also appended comments from the academic editor below, that must be addressed in a revised manuscript. After discussing the paper with the editorial team and an academic editor with relevant expertise, I'm pleased to invite you to revise the paper in response to the reviewers' comments. We plan to send the revised paper to some or all of the original reviewers, and we cannot provide any guarantees at this stage regarding publication.

We ask that you submit your revision by Aug 28 2025 11:59PM. However, if this deadline is not feasible, please contact me by email, and we can discuss a suitable alternative.

Don't hesitate to contact me directly with any questions (afarrell@plos.org).

Best regards,

Alison

Alison Farrell, Ph.D.

Senior Editor

PLOS Medicine

afarrell@plos.org

Comments from the academic editor:

In the current global funding climate and uncertainties about PEPFAR funding continuation it is likely that rate of vertical transmission will go up again, given that ART provision and adherence over pregnancy is likely to be affected negatively. Very early infant diagnosis would thus become more important still.

Further, the diagnosis of non-infection at birth would identify potential transmission risk through breastfeeding (which is now uncommon among infected mothers provided they are on ART), and again this risk may increase if women postpartum are unable to access ART reliably. These issues, should be mentioned in the discussion with an indication that the landscape of vertical transmission is likely to change if PEPFAR funding is withdrawn, in which case identifying infants most at risk is important.

This randomised trial did not compare birth with 6 weeks diagnosis, and only studies whether the birth test adds to the standard of care 6 weeks test. This should be discussed.

Comments from the reviewers:

Reviewer #1: This is a generally well written manuscript. The intention was to undertake a comparative cost and cost-effectiveness study of a point of care early HIV diagnostic test based in two countries (Mozambique and Tanzania) based on a pragmatic cluster-randomised trial.

In the introduction, there should be a brief description of why these two particular countries were chosen.

In the CHEERS checklist it is stated that the HEAP is reported on page 4 of your manuscript but this is incorrect and there is not mention of a HEAP.

There should be justification for why the healthcare system perspective was chosen and why a perspective which also captures out-of-pocket payments was not.

A brief description of why the Abbott and Cepheid technologies were chosen should be provided given that each country only used one of the two machines. What impact is this likely to have on the design, results and conclusions drawn.

In the CHEERS checklist it is stated that the time horizon is reported on page 4 but this is not correct. The time horizon is of particular importance given that the platforms evaluated have an assumed 5-year lifespan. What was the time horizon?

At no point in the manuscript did I see any rationale as to why the study would be undertaken in these two countries using different machines, processes and underlying transmission rates.

Having read the manuscript I am still unsure what to do with the findings - the discussion and conclusions should be framed to allow some recommendation to be made other than more research being undertaken.

Reviewer #2: This is an interesting paper that looks into the costs and cost-effectiveness of POC birth testing. It's an important topic, especially as national programs need to set priorities with limited resources, but could be improved by addressing a few comments.

Major comments

1.Given this is a cost-effectiveness paper, it is unclear why DALYs weren't included as well.

2.It would be very beneficial, given previous cost-effectiveness work on POC, and some of the discussion, for the authors to include a scenario where the devices are shared/integrated with other diseases. There are some results on this included in the discussion, but a clear analysis should be incorporated in the results.

3.It would be helpful to more clearly articulate why the cost of the test is lower for VEID than SOC when the same test is used, per country. Where do those "cost-savings" happen?

4.It is clear and positive that the authors took a slightly different approach to cost-effectiveness by using the health expenditure per capita. However, either include why GDP was not selected more explicitly in the discussion and methods or include that as well.

5.The results could be strengthened further by:

a.The first section and Figure 2 included primary impact data that did not seem part of this cost-effectiveness work, but likely part of the original CID 2024 publication. Instead, this should be shortened or an abbreviated portion put into the methods with a supplementary table as a reference point used, not suggested as new data generated. Likewise, this should be reduced/limited in the discussion.

b.It would be helpful to more clearly note what the costs per test were comprised of. Figure 3 helps; however, distinct dollar values for each portion and how the device was amortized would be helpful.

c.Further, Table 1 is good, but doesn't provide a clear link to the cost per test. It would be helpful here to show how you break down each line item to contribute to the costs identified.

d.Another sensitivity analysis that should be included would be revising the device lifespan to what both countries see as typical/median life spans for the specific devices and/or POC instruments more generally.

e.Further, a final sensitivity analysis would be more clearly showing the transmission rates where POC birth testing becomes cost-effective.

6.The cost per infant should be defined more clearly. In particular, highlighting that it is higher than per test and SOC because they are receiving an additional test. Why are the per infant costs of SOC different from the cost per HIV test? Aren't both only receiving one test and no different cost components are included in the former that weren't in the latter?

7.In the discussion, I would suggest noting that one limitation is the use of POC as SOC and that this is likely to underestimate impact when lab-based testing is used.

8.The authors shied away from providing a clear conclusion to their work. From what it seems, POC birth testing is not at all cost-effective. This is unfortunate, but it is what the data show. Laying that out clearly should be done, not waiting for the last paragraph. However, to this point, a positive spin could be more possible should sensitivity analyses from points 2, 5d, and 5e be completed. Then the authors could more clearly conclude that POC birth testing for all children using specifically purchased devices may not be cost-effective; however, in x, y, z settings this approach is cost-effective. It would provide much clearer conclusions and suggestions for future guidance.

Minor comments

1.The last sentence of the second paragraph in the Abstract is duplicative. Instead some methodology on cost-effectiveness could be included.

2.Lines 222-223: retention in care, adherence to treatment, follow-up HIV testing are not linked or caused by POC EID. This sentence may need revision. Authors should be careful not to suggest this link throughout. Yes, POC can help with linkage to treatment, but thereafter it's about the system and services.

3.Lines 229-231: highlight that all infants testing HIV-negative should receive follow-up testing, not just those with a birth test.

Reviewer #3: This paper presents important cost data on early infant diagnosis. The authors present the challenges well. Clearly earlier identification and treatment of HIV is a public health goal but also a moral obligation. That said, the authors identify the costs associated with early identification. The authors raise the important point that early ART initiation should lead to reduced mortality and the many other steps after diagnosis at birth that need to occur for this to happen- not easy but an important goal. One also wonders about the potential for cure if infants can be started early and appropriately virally suppressed. (https://www-nature-com.libproxy.lib.unc.edu/articles/s41591-024-03105-4)

Some clarifying comments/questions:

Line 135- do reasons for not starting ART vary by arm..same comments for line 137 do reasons for delayed treatment vary by arm- imagine it would?

Line 144- "cost per infant was less than cost per test for SOC due to loss to follow up"- while this is real- it doesn't capture the real world "cost" of a child who is exposed and may have HIV and lost to follow up- it seems sensitivity analyses to address 'survivorship' bias here would be helpful? Inverse probability of treatment weights or imputation? It feels like part of the point of testing at birth is also to capture people while they are at the facility after birth- clearly capturing infections earlier but also capturing people who may not come back which clearly the SOC arm experienced.

Any attachments provided with reviews can be seen via the following link: ********

---

*We ask every co-author listed on the manuscript to fill in a contributing author statement, making sure to declare all competing interests. If any of the co-authors have not filled in the statement, we will remind them to do so when the paper is revised. If all statements are not completed in a timely fashion this could hold up the re-review process. If new competing interests are declared later in the revision process, this may also hold up the submission. Should there be a problem getting one of your co-authors to fill in a statement we will be in contact. Please do not add or remove authors without first discussing this with the handling editor. You can see our competing interests policy here: http://journals.plos.org/plosmedicine/s/competing-interests.

*Please upload any figures associated with your paper as individual TIF or EPS files with 300dpi resolution at resubmission; please read our figure guidelines for more information on our requirements: http://journals.plos.org/plosmedicine/s/figures. While revising your submission, please upload your figure files to the PACE digital diagnostic tool, https://pacev2.apexcovantage.com/. PACE helps ensure that figures meet PLOS requirements. To use PACE, you must first register as a user. Then, login and navigate to the UPLOAD tab, where you will find detailed instructions on how to use the tool. If you encounter any issues or have any questions when using PACE, please email us at PLOSMedicine@plos.org.

*Please ensure that the paper adheres to the PLOS Data Availability Policy (see http://journals.plos.org/plosmedicine/s/data-availability), which requires that all data underlying the study's findings be provided in a repository or as Supporting Information. For data residing with a third party, authors are required to provide instructions with contact information (web or email address) for obtaining the data. Please note that a study author cannot be the contact person for the data. PLOS journals do not allow statements supported by "data not shown" or "unpublished results." For such statements, authors must provide supporting data or cite public sources that include it.

*We expect all researchers with submissions to PLOS in which author-generated code underpins the findings in the manuscript to make all author-generated code available without restrictions upon publication of the work. In cases where code is central to the manuscript, we may require the code to be made available as a condition of publication. Authors are responsible for ensuring that the code is reusable and well documented. Please make any custom code available, either as part of your data deposition or as a supplementary file. Please add a sentence to your data availability statement regarding any code used in the study, e.g. "The code used in the analysis is available from Github [URL] and archived in Zenodo [DOI link]" Please review our guidelines at https://journals.plos.org/plosmedicine/s/materials-software-and-code-sharing and ensure that your code is shared in a way that follows best practice and facilitates reproducibility and reuse. Because Github depositions can be readily changed or deleted, we encourage you to make a permanent DOI'd copy (e.g. in Zenodo) and provide the URL.

*Please confirm that FINANCIAL DISCLOSURES, COI, DAS, AND ETHICS STATEMENTS are correct.

*Please ensure that the study is reported according to the appropriate guideline and include the completed checklist as Supporting Information. When completing the checklist, please use section and paragraph numbers, rather than page numbers. Please add the following statement, or similar, to the Methods: "This study is reported as per [XXXX] guideline (S1 Checklist)."

*Abstract: Please structure your abstract using the PLOS Medicine headings (Background, Methods and Findings, Conclusions). Please combine the Methods and Findings sections into one section.

* Please revise the title as per PLOS Medicine style.

*At this stage, we ask that you include a short, non-technical Author Summary of your research to make findings accessible to a wide audience that includes both scientists and non-scientists. The Author Summary should immediately follow the Abstract in your revised manuscript. This text is subject to editorial change and should be distinct from the scientific abstract. Ideally each sub-heading should contain 2-3 single sentence, concise bullet points containing the most salient points from your study. In the final bullet point of 'What Do These Findings Mean?', please include the main limitations of the study in non-technical language. Please see our author guidelines for more information: https://journals.plos.org/plosmedicine/s/revising-your-manuscript#loc-author-summary.

*Please express the main results with 95% CIs as well as p values. When reporting p values please report as p<0.001 and where higher as the exact p value p=0.002, for example. Throughout, suggest reporting statistical information as follows to improve clarity for the reader "22% (95% CI [13%,28%]; p</=)". Please be sure to define all numerical values at first use.

*Please include page numbers and line numbers in the manuscript file. Use continuous line numbers (do not restart the numbering on each page).

*Please cite the reference numbers in square brackets. Citations should precede punctuation.

FIGURES AND TABLES

*Please provide titles and legends for all figures and tables (including those in Supporting Information files).

*Please define all abbreviations used in each figure/table (including those in Supporting Information files).

*Please consider avoiding the use of red and green in order to make your figure more accessible to those with color blindness.

SUPPLEMENTARY MATERIAL

*Please note that supplementary material will be posted as supplied by the authors. Therefore, please amend it according to the relevant comments outlined here.

*Please cite your Supporting Information as outlined here: https://journals.plos.org/plosmedicine/s/supporting-information

REFERENCES

*PLOS uses the numbered citation (citation-sequence) method and first six authors, et al.

*Please ensure that journal name abbreviations match those found in the National Center for Biotechnology Information (NCBI) databases (http://www.ncbi.nlm.nih.gov/nlmcatalog/journals), and are appropriately formatted and capitalised.

*Where website addresses are cited, please include the complete URL and specify the date of access (e.g. [accessed: 12/06/2023]).

*Please also see https://journals.plos.org/plosmedicine/s/submission-guidelines#loc-references for further details on reference formatting.

HEALTH ECONOMICS / COST-EFFECTIVENESS STUDIES

*Please ensure that the study is reported according to the CHEERS guideline (available from: https://www.equator-network.org/reporting-guidelines/cheers) and include the completed checklist as Supporting Information. Please add the following statement, or similar, to the Methods: "This study is reported as per the Strengthening the Consolidated Health Economic Evaluation Reporting Standards 2022 (CHEERS 2022) Statement (S1 Checklist)." When completing the checklist, please use section and paragraph numbers, rather than page numbers.

---

## [Decision Letter · Decision Letter 2]

8 Dec 2025

Dear Dr Elsbernd,

Many thanks for submitting your manuscript "Comparative cost and cost-effectiveness of point-of-care early infant HIV diagnosis at birth: Findings from a pragmatic cluster-randomized trial in Mozambique and Tanzania" (PMEDICINE-D-25-00626R2) to PLOS Medicine. The paper has been reviewed by subject experts and a statistician; their comments are included below and can also be accessed here: [LINK]

As you will see, the reviewer 2 is now satisfied with the revisions to the manuscript. Unfortunately, we did not receive further comments from the other reviewers and so sought input from a member of the PLOS Medicine statistical reviewer board with expertise in cost effectiveness analysis. As you will see below, this reviewer has raised a number of concerns requiring clarification and justification of analytical decisions, with ensuing revision to the manuscript. We also require that you extend your discussion of study limitations to address the concerns raised. After discussing the paper with the editorial team and an academic editor with relevant expertise, I'm pleased to invite you to revise the paper in response to the reviewers' comments. We plan to send the revised paper to some or all of the original reviewers, and we cannot provide any guarantees at this stage regarding publication.

Please note the formatting requirements for revising manuscripts: https://journals.plos.org/plosmedicine/s/revising-your-manuscript#loc-author-summary.

* Please confirm that your title complies with PLOS Medicine's style. Your title must be nondeclarative and not a question. It should begin with main concept if possible. "Effect of" should be used only if causality can be inferred, i.e., for an RCT. Please place the study design ("A randomized controlled trial," "A retrospective study," "A modelling study," etc.) in the subtitle (ie, after a colon).

* In the last bullet point of the Author study, please add the study limitations in very general language.

* In figures 3 and 6, please change one of the blue colours in the figures for improved contrast.

We ask that you submit your revision by Dec 29 2025 11:59PM. However, if this deadline is not feasible, please contact me by email, and we can discuss a suitable alternative.

Don't hesitate to contact me directly with any questions (afarrell@plos.org).

Best regards,

Alison

Alison Farrell, Ph.D.

Senior Editor

PLOS Medicine

afarrell@plos.org

Comments from the academic editor:

The aim of the paper could be made more clear--i.e. to look at birth plus 4-6 week testing versus 4-6 week testing only, and that thus the used time horizon is appropriate as any assumptions after 6 weeks would be the same in both arms. There is interest in knowing what the addition of a birth HIV tests adds to getting HIV infected infants on ART as soon as feasible and thus reduce mortality - all the more important now, with reduced funding and consequent impact on health care/systems, when pregnant women may have reduced HIV testing and ART uptake and thus more likely to have vertically infected children.

Comments from the reviewers:

Reviewer #2: Thank you for the thorough and robust consideration of all author comments and incorporation into the final manuscript. Congratulations!

Reviewer #4: Major Comments

1.The study's outcomes (ART initiation and EID uptake) are intermediate process indicators rather than final health outcomes. The authors should clarify that this represents a cost-consequence or cost-effectiveness analysis using intermediate outcomes, not a full cost-utility or cost-effectiveness analysis based on health outcomes (e.g., DALYs or QALYs).

2.The study applies empirically derived country-level thresholds and GDP per capita values as comparators. However, these thresholds are expressed as cost per life-year gained, not per ART initiation. This mismatch should be explicitly acknowledged, as it limits direct interpretability of the ICERs presented.

3.While "ART initiation within one week" is programmatically important, its link to health benefit is indirect. Without estimating life-years gained, ICERs per early ART initiation are difficult to interpret. The threshold analysis is creative but cannot substitute for a formal utility-based analysis.

4.The manuscript reports ICERs (e.g., ~$3,000 per additional early ART initiation) and subsequently infers that 16-28 life-years gained per early initiation would be required for cost-effectiveness. This calculation reflects a reverse application of cost-effectiveness thresholds rather than a modelled ICER per life-year gained. The authors should clearly state that this is a threshold-based calculation, not a modeled estimate, and present the formula linking incremental cost per ART initiation to the implied cost per life-year gained.

5.The 12-week analytic horizon is too short to capture meaningful health and cost outcomes of early ART. Although the authors estimate the required life-years for cost-effectiveness, this remains a post hoc conceptual exercise rather than a formal model. The limitations of this short horizon should be discussed explicitly, and the potential value of a decision-analytic or Markov extrapolation acknowledged.

6.The study adopts a healthcare provider perspective, excluding patient and societal costs. The statement that patient costs were "minimal and non-monetary" lacks empirical justification. Please provide evidence or references supporting this assumption, or discuss its limitations.

7.The exclusion of confirmatory central laboratory testing and potential shared resource allocation may underestimate total costs. The method for apportioning shared costs (e.g., electricity, space, analyzer use for other assays) should be described more clearly.

8.Details of the Bayesian hierarchical (hurdle Gamma) model, priors, and convergence diagnostics are only provided in the supplement. A concise summary of the model structure, priors, and diagnostics (R� values, trace plots, posterior predictive checks) should be included in the main text for transparency.

9.The reported posterior correlation (ρ = −0.25) is not interpreted. The authors should clarify what this correlation indicates about the relationship between costs and effects.

10.The classification of zero-cost cases (infants lost to follow-up) as structural zeros should be explicitly justified, as this may bias mean cost estimates downward.

11.The manuscript does not specify whether intra-cluster correlation (ICC) was accounted for in bootstrapping or Bayesian estimation. ICCs for both cost and effect outcomes should be reported, and the handling of clustering in ICER estimation should be clarified.

12.The use of empirically derived thresholds is appropriate, but the link between these thresholds and local opportunity costs in Mozambique and Tanzania should be better justified. A short discussion on their applicability for decision-makers would strengthen the policy relevance.

Minor Comments

13.The probabilistic sensitivity analysis (PSA) lacks detail regarding which parameters were varied (e.g., costs, clinical inputs). Please clarify.

14.The manuscript states that costs were converted to 2020 USD; please detail the inflation adjustment process and indices used (2020-2021).

15.Include exchange rate and inflation details in the main text rather than only in the supplement.

16.Clearly define the numerator and denominator of the ICERs (e.g., per HIV-exposed or HIV-positive infant).

17.Ensure consistency between the text and CHEERS checklist—some items (e.g., 12 and 17) lack sufficient detail in the manuscript.

18.Given the discussion around high-risk infants, consider addressing equity or distributional aspects of cost-effectiveness.

19.Maintain consistent terminology (e.g., use "95% CrI" for Bayesian intervals and "incremental cost per additional infant initiating ART within 1 week" throughout).

20.The joint Bayesian model combines cost and effect outcomes, but diagnostics (e.g., posterior predictive checks, residual plots) are only briefly mentioned in the supplement. Include a short summary of model validation.

21.The deterministic sensitivity analysis ranges (Figure 6) appear arbitrary—indicate whether these are empirical, elicited, or assumed (e.g., ±25%).

22.The probabilistic framework does not appear to propagate uncertainty from all cost inputs. Broader PSA incorporating input uncertainty (e.g., consumables, lifespan, capital costs) would be expected in a full CEA.

23.The CEACs (Figure 4) should explicitly label axes and clarify their relationship to willingness-to-pay thresholds.

24.The use of p-values (e.g., p=0.112, p<0.0001) is inconsistent with a Bayesian analysis. Either justify the hybrid frequentist approach or replace these with Bayesian credible intervals.

25.For clarity, replace "ICER per 1-week ART initiation" with "incremental cost per additional infant initiating ART within 1 week."

---

*We ask every co-author listed on the manuscript to fill in a contributing author statement, making sure to declare all competing interests. If any of the co-authors have not filled in the statement, we will remind them to do so when the paper is revised. If all statements are not completed in a timely fashion this could hold up the re-review process. If new competing interests are declared later in the revision process, this may also hold up the submission. Should there be a problem getting one of your co-authors to fill in a statement we will be in contact. Please do not add or remove authors without first discussing this with the handling editor. You can see our competing interests policy here: http://journals.plos.org/plosmedicine/s/competing-interests.

*Please upload any figures associated with your paper as individual TIF or EPS files with 300dpi resolution at resubmission; please read our figure guidelines for more information on our requirements: http://journals.plos.org/plosmedicine/s/figures. While revising your submission, we strongly recommend that you use PLOS's NAAS tool (https://ngplosjournals.pagemajik.ai/artanalysis) to test your figure files. NAAS can convert your figure files to the TIFF file type and meet basic requirements (such as print size, resolution), or provide you with a report on issues that do not meet our requirements and that NAAS cannot fix.

After uploading your figures to PLOS's NAAS tool - https://ngplosjournals.pagemajik.ai/artanalysis, NAAS will process the files provided and display the results in the "Uploaded Files" section of the page as the processing is complete.

If the uploaded figures meet our requirements (or NAAS is able to fix the files to meet our requirements), the figure will be marked as "fixed" above. If NAAS is unable to fix the files, a red "failed" label will appear above.

When NAAS has confirmed that the figure files meet our requirements, please download the file via the download option, and include these NAAS processed figure files when submitting your revised manuscript.

*Please ensure that the paper adheres to the PLOS Data Availability Policy (see http://journals.plos.org/plosmedicine/s/data-availability), which requires that all data underlying the study's findings be provided in a repository or as Supporting Information. For data residing with a third party, authors are required to provide instructions with contact information (web or email address) for obtaining the data. Please note that a study author cannot be the contact person for the data. PLOS journals do not allow statements supported by "data not shown" or "unpublished results." For such statements, authors must provide supporting data or cite public sources that include it.

*We expect all researchers with submissions to PLOS in which author-generated code underpins the findings in the manuscript to make all author-generated code available without restrictions upon publication of the work. In cases where code is central to the manuscript, we may require the code to be made available as a condition of publication. Authors are responsible for ensuring that the code is reusable and well documented. Please make any custom code available, either as part of your data deposition or as a supplementary file. Please add a sentence to your data availability statement regarding any code used in the study, e.g. "The code used in the analysis is available from Github [URL] and archived in Zenodo [DOI link]" Please review our guidelines at https://journals.plos.org/plosmedicine/s/materials-software-and-code-sharing and ensure that your code is shared in a way that follows best practice and facilitates reproducibility and reuse. Because Github depositions can be readily changed or deleted, we encourage you to make a permanent DOI'd copy (e.g. in Zenodo) and provide the URL.

*[EDITOR: CHECK FINANCIAL DISCLOSURES, COI, DAS, AND ETHICS STATEMENTS AND INCLUDE ANY NECESSARY REQUESTS]

*Please ensure that the study is reported according to the [XXXX] guideline and include the completed [XXXX] checklist as Supporting Information. When completing the checklist, please use section and paragraph numbers, rather than page numbers. Please add the following statement, or similar, to the Methods: "This study is reported as per [XXXX] guideline (S1 Checklist)."

*Abstract: Please structure your abstract using the PLOS Medicine headings (Background, Methods and Findings, Conclusions). Please combine the Methods and Findings sections into one section.

*At this stage, we ask that you include a short, non-technical Author Summary of your research to make findings accessible to a wide audience that includes both scientists and non-scientists. The Author Summary should immediately follow the Abstract in your revised manuscript. This text is subject to editorial change and should be distinct from the scientific abstract. Ideally each sub-heading should contain 2-3 single sentence, concise bullet points containing the most salient points from your study. In the final bullet point of 'What Do These Findings Mean?', please include the main limitations of the study in non-technical language. Please see our author guidelines for more information: https://journals.plos.org/plosmedicine/s/revising-your-manuscript#loc-author-summary.

*Please express the main results with 95% CIs as well as p values. When reporting p values please report as p<0.001 and where higher as the exact p value p=0.002, for example. Throughout, suggest reporting statistical information as follows to improve clarity for the reader "22% (95% CI [13%,28%]; p</=)". Please be sure to define all numerical values at first use.

*Please include page numbers and line numbers in the manuscript file. Use continuous line numbers (do not restart the numbering on each page).

*Please cite the reference numbers in square brackets. Citations should precede punctuation.

FIGURES AND TABLES

*Please provide titles and legends for all figures and tables (including those in Supporting Information files).

*Please define all abbreviations used in each figure/table (including those in Supporting Information files).

*Please consider avoiding the use of red and green in order to make your figure more accessible to those with color blindness.

SUPPLEMENTARY MATERIAL

*Please note that supplementary material will be posted as supplied by the authors. Therefore, please amend it according to the relevant comments outlined here.

*Please cite your Supporting Information as outlined here: https://journals.plos.org/plosmedicine/s/supporting-information

REFERENCES

*PLOS uses the numbered citation (citation-sequence) method and first six authors, et al.

*Please ensure that journal name abbreviations match those found in the National Center for Biotechnology Information (NCBI) databases (http://www.ncbi.nlm.nih.gov/nlmcatalog/journals), and are appropriately formatted and capitalised.

*Where website addresses are cited, please include the complete URL and specify the date of access (e.g. [accessed: 12/06/2023]).

*Please also see https://journals.plos.org/plosmedicine/s/submission-guidelines#loc-references for further details on reference formatting.

[STUDY TYPE-SPECIFIC REQUESTS - DELETE SECTIONS AS NECESSARY]

RCTs [REFER TO RCT CHECKLIST AND MEETING NOTES FOR DETAILS TO ADD]

*PLOS Medicine requires that all trials be prospectively registered in one of registries recognized by WHO. Please ensure that study registration details are included in the Methods section.

*Please structure the Methods section using the following sub-headings: Study design and participants, Randomization and masking, Procedures, Outcomes, Statistical analysis.

*The following outcomes measures [ADD DETAILS AS NEEDED OR DELETE BULLET POINT] appear to differ between the submitted manuscript and the protocol [and/or trial registry]. Please clarify and explain all discrepancies between the paper and protocol. If the outcomes were not prespecified in the protocol, please define them in the Methods (Outcomes section) as post hoc and explain why they were added. Post-hoc comparisons should be presented as hypothesis generating rather than conclusive.

*Please ensure that all prespecified outcomes (primary, secondary, and exploratory) are listed in the Methods/Outcomes section and indicate whether there are outcomes that are not presented in the current report.

*Please specify the dates (Month Day, Year) during which study enrollment and follow up occurred.

*Please include absolute numbers wherever you report percentages; eg, n/N (%)

*Please present the safety data for the study including numbers of specific events and whether or not adverse events are thought to be related to treatment. AEs should be reported in the abstract, per CONSORT and CONSORT-Harms.

*Please complete the CONSORT checklist (https://www.equator-network.org/reporting-guidelines/consort/) and ensure that all components of CONSORT are present in the manuscript, including how randomization was performed, allocation concealment, blinding of intervention, definition of lost to follow-up, power statement. When completing the checklist, please use section and paragraph numbers, rather than page numbers.

*Please report your abstract according to CONSORT for abstracts, following the PLOS Medicine abstract structure (Background, Methods and Findings, Conclusions) https://www.equator-network.org/reporting-guidelines/consort-abstracts/

*If your trial had to undergo important modifications in response to extenuating circumstances, please complete the CONSERVE-CONSORT checklist and provide in your Supporting Information; (https://www.equator-network.org/reporting-guidelines/guidelines-for-reporting-trial-protocols-and-completed-trials-modified-due-to-the-covid-19-pandemic-and-other-extenuating-circumstances-the-conserve-2021-statement/). When completing the checklist, please use section and paragraph numbers, rather than page numbers.

*In keeping with our commitment to Open Science, please include the study protocol document and analysis plan (including any amendments) as Supporting Information to be published with the manuscript if accepted.

*Please note that PLOS Medicine requires prospective, public registration of a data sharing plan (as part of mandatory clinical trials registration) for all clinical trials that began enrollment on or after January 1, 2019, in accordance with ICMJE requirements.

OBSERVATIONAL STUDIES

*Abstract: Please include the study design, population and setting, number of participants, years during which the study took place (enrollment and follow up), length of follow up, and main outcome measures.

*Please ensure that the study is reported according to the STROBE (or appropriate STOBE extension) guideline (available from: https://www.equator-network.org/reporting-guidelines/strobe) and include the completed STROBE (or STROBE extension) checklist as Supporting Information. Please add the following statement, or similar, to the Methods: "This study is reported as per the Strengthening the Reporting of Observational Studies in Epidemiology (STROBE) guideline (S1 Checklist)." When completing the checklist, please use section and paragraph numbers, rather than page numbers.

*[FOR POPULATION HEALTH/REGISTRY STUDIES] Please ensure that the study is reported according to the RECORD guideline (available from https://www.record-statement.org) and include the completed checklist as Supporting Information. Please add the following statement, or similar, to the Methods: "This study is reported as per the Reporting of Studies Conducted using Observational Routinely-Collected Data (RECORD) guideline (S1 Checklist)." When completing the checklist, please use section and paragraph numbers, rather than page numbers.

*[FOR POPULATION HEALTH ESTIMATES] Please ensure that the study is reported according to the GATHER statement (available from https://www.equator-network.org/reporting-guidelines/gather-statement) and include the completed checklist as Supporting Information. Please add the following statement, or similar, to the Methods: "This study is reported as per the Guidelines for Accurate and Transparent Health Estimates Reporting (GATHER) statement (S1 Checklist)." When completing the checklist, please use section and paragraph numbers, rather than page numbers.

*[FOR MEDIATION ANALYSES] We recommend that the study is reported according to the AGReMA statement (https://agrema-statement.org/#:~:text=AGReMA%20is%20an%20evidence%2D%20and,randomised%20trials%20and%20observational%20studies) and include the completed checklist as Supporting Information. Please add the following statement, or similar, to the Methods: "This study is reported as per the Guideline for Reporting Mediation Analyses (AGReMA) statement (S1 Checklist)." When completing the checklist, please use section and paragraph numbers, rather than page numbers.

*For all observational studies, in the manuscript text, please indicate: (1) the specific hypotheses you intended to test, (2) the analytical methods by which you planned to test them, (3) the analyses you actually performed, and (4) when reported analyses differ from those that were planned, transparent explanations for differences that affect the reliability of the study's results. If a reported analysis was performed based on an interesting but unanticipated pattern in the data, please be clear that the analysis was data driven.

*Please state in the Methods section whether the study had a prospective protocol or analysis plan. If a prospective analysis plan (from your funding proposal, IRB or other ethics committee submission, study protocol, or other planning document written before analyzing the data) was used in designing the study, please include the relevant document(s) with your revised manuscript as a Supporting Information file to be published alongside your study and cite it in the Methods section. A legend for this file should be included at the end of your manuscript. If no such document exists, please make sure that the Methods section transparently describes when analyses were planned, and when/why any data-driven changes to analyses took place. Changes in the analysis, including those made in response to peer review comments, should be identified as such in the Methods section of the paper, with rationale.

MODELLING STUDIES

The following list is derived from Geoffrey P Garnett, Simon Cousens, Timothy B Hallett, Richard Steketee, Neff Walker. Mathematical models in the evaluation of health programmes. (2011) Lancet DOI:10.1016/S0140-6736(10)61505-X:

*If pertinent, please provide a diagram that shows the model structure, including how the natural history of the disease is represented, the process and determinants of disease acquisition, and how the putative intervention could affect the system.

*Please provide a complete list of model parameters, including clear and precise descriptions of the meaning of each parameter, together with the values or ranges for each, with justification or the primary source cited and important caveats about the use of these values noted.

*Please provide a clear statement about how the model was fitted to the data, including goodness-of-fit measure, the numerical algorithm used, which parameter varied, constraints imposed on parameter values, and starting conditions.

*For uncertainty analyses, please state the sources of uncertainties quantified and not quantified [can include parameter, data, and model structure].

*Please provide sensitivity analyses to identify which parameter values are most important in the model. Uncertainty estimates seek to derive a range of credible results on the basis of an exploration of the range of reasonable parameter values. The choice of method should be presented and justified.

*Please discuss the scientific rationale for the choice of model structure and identify points where this choice could influence conclusions drawn. Please also describe the strength of the scientific basis underlying the key model assumptions.

*For studies that develop a prediction model or evaluate its performance, please ensure that the study is reported according to the TRIPOD statement (https://www.equator-network.org/reporting-guidelines/tripod-statement) and include the completed checklist as Supporting Information. Please add the following statement, or similar, to the Methods: "This study is reported as per the Transparent Reporting of a Multivariable Prediction Model for Individual Prognosis Or Diagnosis (TRIPOD) statement (S1 Checklist)." For studies using machine learning, please use the TRIPOD-AI checklist. When completing the checklist, please use section and paragraph numbers, rather than page numbers.

DIAGNOSTIC STUDIES

*Please ensure that the study is reported according to the STARD guideline (https://www.equator-network.org/reporting-guidelines/stard/) and include the completed STARD checklist as Supporting Information. Please add the following statement, or similar, to the Methods: "This study is reported as per the Standards for Reporting of Diagnostic Accuracy (STARD) guideline (S1 Checklist)." When completing the checklist, please use section and paragraph numbers, rather than page numbers.

*Please structure your Abstract according to STARD for Abstracts (https://www.equator-network.org/reporting-guidelines/stard-abstracts/).

*Please structure the Methods section using the following sub-headings: Study design, Participants, Test methods, Analysis.

*Please include a diagram to describe the flow of participants through the study (typically figure 1).

MENDELIAN RANDOMIZATION STUDIES

*Please ensure that the study is reported according to the STROBE-MR guideline (https://www.equator-network.org/reporting-guidelines/strobe/) and include the completed STROBE-MR checklist as Supporting Information. Please add the following statement, or similar, to the Methods: "This study is reported as per the Strengthening the Reporting of Observational Studies in Epidemiology (STROBE) guideline, specific for mendelian randomization (S1 Checklist)." When completing the checklist, please use section and paragraph numbers, rather than page numbers.

*In the Introduction, please describe the exposure and the evidence for a potential causal relationship between exposure and outcome.

*In the Methods, please explicitly state the 3 core instrumental variable assumptions for the main analysis (relevance, independence, and exclusion restriction), as well assumptions for any additional or sensitivity analysis.

*In the Methods, please describe the MR estimator (e.g., 2-stage least squares, Wald ratio) and related statistics. Detail the included covariates and, in case of 2-sample MR, whether the same covariate set was used for adjustment in the 2 samples.

*If you are presenting an instrumental variable estimate, please compare this to the conventional observational estimate.

*Report the associations between genetic variant and exposure and between genetic variant and outcome, preferably on an interpretable scale.

*Report MR estimates of the relationship between exposure and outcome and the measures of uncertainty from the MR analysis, on an interpretable scale, such as odds ratio or relative risk per SD difference.

*If relevant, please consider translating estimates of relative risk into absolute risk for a meaningful time period.

*Please consider including plots to visualize results (e.g., forest plot, scatterplot of associations between genetic variants and outcome vs between genetic variants and exposure).

SURVEY-BASED STUDIES

*Please ensure that the study is reported according to the CROSS guideline (https://www.equator-network.org/reporting-guidelines/a-consensus-based-checklist-for-reporting-of-survey-studies-cross/) and include the completed CROSS checklist as Supporting Information. Please add the following statement, or similar, to the Methods: "This study is reported as per A Consensus-Based Checklist for Reporting of Survey Studies (CROSS) guideline (S1 Checklist)." When completing the checklist, please use section and paragraph numbers, rather than page numbers.

*Please report your survey response rates according to AAPOR recommendations (https://aapor.org/standards-and-ethics/best-practices/)

*Please define how the population surveyed was sampled.

*Please compare characteristics of respondents and nonrespondents if possible.

*If sequential waves of the survey were sent, please specify whether the characteristics of respondents changed over time or remained constant.

*Please include the survey response rate in the Abstract.

*Please include a copy of the survey in the supplementary files.

SYSTEMATIC REVIEWS & META-ANALYSES

*Please report your SR/MA according to the PRISMA guidelines provided at the EQUATOR site. http://www.equator-network.org/reporting-guidelines/prisma/. Please provide the completed PRISMA checklist as Supporting Information. When completing the checklist, please use section and paragraph numbers, rather than page numbers. Please add the following statement, or similar, to the Methods: "This study is reported as per the Preferred Reporting Items for Systematic Reviews and Meta-Analyses (PRISMA) guideline (S1 Checklist)."

*Abstract: Please report your abstract according to PRISMA for abstracts (https://doi.org/10.1371/journal.pmed.1001419) following the PLOS Medicine abstract structure (Background, Methods and Findings, Conclusions). Please ensure you provide dates of search, data sources, number of studies included, types of study designs included, eligibility criteria, and synthesis/appraisal methods.

*Please note that we expect searches to be updated to within 6 months of the time of submission.

QUALITATIVE STUDIES

*Please report your qualitative study according to the appropriate study design provided at (http://www.equator-network.org/?post_type=eq_guidelines&eq_guidelines_study_design=qualitative-research&eq_guidelines_clinical_specialty=0&eq_guidelines_report_section=0&s=) and provide the relevant completed checklist as a supplemental file. In the checklist, please include sufficient text excerpted from the manuscript to explain how you accomplished all applicable items. When completing checklists, please use section and paragraph numbers, rather than page numbers.

*We recommend that authors use the COREQ checklist, or other relevant checklists listed by the Equator Network, such as the SRQR, to ensure complete reporting (see: http://www.equator-network.org/?post_type=eq_guidelines&eq_guidelines_study_design=qualitative-research&eq_guidelines_clinical_specialty=0&eq_guidelines_report_section=0&s=). Please add the following statement, or similar, to the Methods: "This study is reported as per the Consolidated criteria for reporting qualitative research (COREQ): a 32-item checklist for interviews and focus groups (S1 Checklist)."

*In general, we expect qualitative studies to include the following: 1) defined objectives or research questions; 2) description of the sampling strategy, including rationale for the recruitment method, participant inclusion/exclusion criteria and the number of participants recruited; 3) detailed reporting of the data collection procedures; 4) data analysis procedures described in sufficient detail to enable replication; 5) a discussion of potential sources of bias; and 6) a discussion of limitations.

HEALTH ECONOMICS / COST-EFFECTIVENESS STUDIES

*Please ensure that the study is reported according to the CHEERS guideline (available from: https://www.equator-network.org/reporting-guidelines/cheers) and include the completed checklist as Supporting Information. Please add the following statement, or similar, to the Methods: "This study is reported as per the Strengthening the Consolidated Health Economic Evaluation Reporting Standards 2022 (CHEERS 2022) Statement (S1 Checklist)." When completing the checklist, please use section and paragraph numbers, rather than page numbers.

---

## [Decision Letter · Decision Letter 3]

3 Mar 2026

Dear Dr. Elsbernd,

Thank you very much for re-submitting your manuscript "Comparative cost and cost-effectiveness of point-of-care early infant HIV diagnosis at birth: Findings from a pragmatic cluster-randomized trial in Mozambique and Tanzania" (PMEDICINE-D-25-00626R3) for review by PLOS Medicine.

I have discussed the paper with my colleagues and the academic editor, and it was also seen again by the statistical reviewer who has some remaining requests. I am pleased to say that provided the remaining editorial and reviewer concerns and production issues are dealt with, we are planning to accept the paper for publication in the journal.

We ask that you include the sensitivity analyses requested by the statistical reviewer in point 1, and address the other issues raised in the manuscript text.

[LINK]

We look forward to receiving the revised manuscript by Mar 10 2026 11:59PM.

Sincerely,

Alison Farrell, Ph.D.

Senior Editor

PLOS Medicine

plosmedicine.org

Requests from Editors:

Title:

* Please confirm that your title complies with PLOS Medicine's style. Your title must be nondeclarative and not a question. It should begin with main concept if possible. "Effect of" should be used only if causality can be inferred, i.e., for an RCT. Please place the study design ("A randomized controlled trial," "A retrospective study," "A modelling study," etc.) in the subtitle (ie, after a colon). We suggest: “Point-of-care early infant HIV diagnosis at birth in a pragmatic cluster-randomized trial in Mozambique and Tanzania: a comparative cost and cost-effectiveness study”

Abstract:

* Please confirm that your abstract complies with our requirements, including format (three sections: Background, Methods and Findings, and Conclusions) and providing all the information relevant to this study type https://journals.plos.org/plosmedicine/s/submission-guidelines#loc-abstract.

Please clarify in the Abstract whether this was a prespecified secondary analysis.

Author Summary:

* In the author summary, in the final bullet point of 'What Do These Findings Mean?', please include the main limitations of the study in non-technical language.

The second section of the Author summary should be titled “What did the researchers do and find?”

The text of the Author Summary is subject to editorial change and should be distinct from the scientific abstract. In the final bullet point of ‘What Do These Findings Mean?’ Please include the main limitations of the study in non-technical language.

Please see our author guidelines for more information: https://journals.plos.org/plosmedicine/s/revising-your-manuscript#loc-author-summary.

Main text:

* Please ensure that the Introduction ends with a clear description of the study question or hypothesis.

* Please ensure that all abbreviations are defined at first use throughout the text.

* Please confirm that all numbers presented in the abstract are present and identical to numbers presented in the main manuscript text.

* Please revise for use of patient-centered language. Please note that patient-centered language is constructed with the use of post-modified nouns (e.g. 'patients with psoriasis’ (or similar) instead of ‘psoriasis patients’) putting the person first in the sentence structure.

* Statistical reporting: Please revise throughout the manuscript, including tables and figures.

- Please report statistical information as follows to improve clarity for the reader ""22% (95% CI [13,28]; p</=)"". CI ranges in your manuscript need to expressed with commas, not hyphens.

- Please separate upper and lower bounds with commas instead of hyphens as the latter can be confused with reporting of negative values.

- Please repeat statistical definitions (HR, CI etc.) for each set of parentheses.

Figures:

* Please define all elements of box plots in the figure caption - center line, box limits and whiskers.

* Please convert any stacked bar charts to another data representation for example a table, or other type of graph.

Figures 3 and S2 contain stacked bar graphs. If these cannot be converted to another format, we require that you add the underlying data to a table for each figure in the supplementary material.

Discussion:

* Please remove the 'conclusions' subheading from the discussion.

Acknowledgments:

* Please include an Acknowledgments section in your manuscript that recognizes study participants and individuals who played a role in data collection or participant care or involvement.

Disclosures:

*Please revise the COI statement in view of the fact that one of the authors (TB) is editor-in-chief of PLOS Medicine. For example, “TB is editor-in-chief of PLOS Medicine. The other authors declare no competing interests.

* The funding statement should include: specific grant numbers, initials of authors who received each award, URLs to sponsors’ websites. Also, please state whether any sponsors or funders (other than the named authors) played any role in study design, data collection and analysis, the decision to publish, or preparation of the manuscript.

* It appears that one or more study authors is affiliated with one or more of the agencies that funded the study. Thus, the statement “The funders had no role in study design, data collection and analysis, decision to publish, or preparation of the manuscript” does not apply. Please revise the Financial Disclosure accordingly, as in "[Author name] is [author's role] at [funding agency]. The funders had no other role in study design…..”

* All authors must declare their relevant competing interests per the PLOS policy, which can be seen here: https://journals.plos.org/plosmedicine/s/competing-interests For authors with ties to industry, please indicate whether any of the interests has a financial stake in the results of the current study.

* Please report your economic analysis according to the appropriate study design provided at http://www.equator-network.org/?post_type=eq_guidelines&eq_guidelines_study_design=economic-evaluations&eq_guidelines_clinical_specialty=0&eq_guidelines_report_section=0&s= and provide the relevant completed checklist. In the checklist please include sufficient text excerpted from the manuscript to explain how you accomplished all applicable items.

* Please use sections rather than page numbers or lines in the checklist.

* PLOS defines the “minimal data set” to consist of the data set used to reach the conclusions drawn in the manuscript with related metadata and methods, and any additional data required to replicate the reported study findings in their entirety. Authors do not need to submit their entire data set, or the raw data collected during an investigation. Please submit the following data:

The values behind the means, standard deviations and other measures reported;

The values used to build graphs;

The points extracted from images for analysis.

* Please confirm that the data in the manuscript adheres to the minimal data set requirements.

* The Data Availability Statement (DAS) requires revision given that the IPD is not included in the manuscript files. For each data source used in your study:

* For studies in which a novel model is central to the manuscript's findings, authors are responsible for providing the source code needed to replicate the study's findings in a repository (such as GitHub, SourceForge or Bitbucket) or a cloud computing service (such as Code Ocean). Protection of authors’ intellectual property will not be cause for exception. Please explain in the manuscript’s Data Availability Statement how readers can access the shared code.

Comments from Reviewers:

Reviewer #4: Major comments

1) Although the informative priors on the study arm effects (Normal(-5,1) and Normal(5,1)) are justified conceptually, no formal prior sensitivity analysis is presented. Given the strength of these priors and the relatively small number of HIV-positive infants, I recommend conducting sensitivity analyses using weaker priors (e.g., Normal(0,2.5) or similar) to demonstrate that posterior inferences are robust and not materially driven by prior assumptions.

2) It is stated that no random effects were included in the 1-week ART initiation model. Given the cluster-randomised design and observed clustering in other outcomes, please clarify the rationale for excluding site-level random effects in this model. If the decision was based on negligible ICC, convergence issues, or limited events, this should be explicitly stated and supported.

3) The ICER is calculated as the ratio of mean incremental cost to mean incremental effect. Please clarify whether ICERs were also evaluated at the posterior draw level (i.e., ΔCᵢ/ΔEᵢ) and whether the chosen approach preserves the joint uncertainty structure between cost and effectiveness. If not, please justify the methodological choice.

4) The choice of a hurdle Gamma model for costs is reasonable, but the manuscript does not report whether alternative specifications (e.g., log-normal, zero-inflated Gamma) were considered. If model comparison was conducted (e.g., LOO-IC or WAIC), please report results; otherwise, please justify the final specification more explicitly.

5) The redistribution of HIV testing costs to HIV-positive infants relies on bootstrap-derived intrauterine transmission probabilities. While this approach is transparent, additional clarification on the epidemiological plausibility of the 0-40% sensitivity range would strengthen the robustness of this assumption.

6) The decision to benchmark required life-years gained against CHER trial results assumes transferability across settings and time periods. Please expand discussion of structural uncertainty related to differences in ART regimens, background mortality, adherence, and health system context.

Minor comments

7) Please clarify explicitly the analytic perspective (e.g., health system perspective) and confirm that patient-incurred or societal costs were intentionally excluded.

8) The rationale for limiting ART cost inclusion to 4 weeks (Mozambique) and 6 weeks (Tanzania) should be more clearly justified, including discussion of how this conservative assumption affects ICER interpretation.

9) Capital cost lifespan assumptions (5-year platform, 10-year facility upgrades) materially influence annualized fixed costs. Consider briefly discussing how sensitive results are to these assumptions.

10) Since PoC platforms were shared with other services (particularly in Tanzania), please clarify how overhead and shared usage were allocated and whether alternative allocation rules were explored.

11) It would be helpful to indicate whether prior-posterior overlap or prior predictive checks were examined for key parameters to demonstrate that inference is primarily data-driven.

12) Consider briefly discussing structural uncertainty (e.g., alternative modelling frameworks or longer-term extrapolation approaches), even if not implemented.

[LINK]

---

## [Decision Letter · Decision Letter 4]

8 Apr 2026

Dear Dr. Elsbernd,

Thank you very much for re-submitting your manuscript "Point-of-care early infant HIV diagnosis at birth in a pragmatic cluster-randomized trial in Mozambique and Tanzania: a comparative cost and cost-effectiveness study" (PMEDICINE-D-25-00626R4) for review by PLOS Medicine.

I have discussed the paper with my colleagues and the academic editor and it was also seen again by one reviewer, whose remaining comments are below. I am pleased to say that provided the remaining reviewer concerns are addressed as per our recent correspondence, and the editorial and production issues are dealt with, we are planning to accept the paper for publication in the journal.

Please confirm or revise the title.

Please note that the last sentence of the Author Summary is not grammatically correct and needs to be revised.

We also require the following authors to complete the questionnaire with the COI declaration: AM, NEN, SB, RE.

[LINK]

We look forward to receiving the revised manuscript by Apr 15 2026 11:59PM.

Sincerely,

Alison Farrell, Ph.D.

Senior Editor

PLOS Medicine

plosmedicine.org

Requests from Editors:

HEALTH ECONOMICS / COST-EFFECTIVENESS STUDIES

* Please ensure that the study is reported according to the CHEERS guideline (available from: https://www.equator-network.org/reporting-guidelines/cheers) and include the completed checklist as Supporting Information. Please add the following statement, or similar, to the Methods: "This study is reported as per the Strengthening the Consolidated Health Economic Evaluation Reporting Standards 2022 (CHEERS 2022) Statement (S1 Checklist)." When completing the checklist, please use section and paragraph numbers, rather than page numbers.

Comments from Reviewers:

Reviewer #4: I thank the authors for their detailed revisions and for improving the clarity of the statistical and costing components of the analysis. The manuscript has been strengthened in several respects. However, one important methodological issue remains insufficiently resolved, which affects the interpretation of the economic evaluation.

Major comment:

1) The analysis defines cost-effectiveness in terms of incremental cost per additional infant initiating ART within one week, which is an intermediate (process) outcome. However, the results are subsequently interpreted using cost-effectiveness thresholds expressed as cost per life-year gained. This represents a mismatch in outcome units, as cost-effectiveness thresholds are only interpretable when applied to ICERs expressed in the same health outcome units (e.g., cost per life-year or QALY gained), as established in standard economic evaluation methodology. While the authors attempt to address this by estimating the number of life-years that would need to be gained per additional early ART initiation, this constitutes a threshold (break-even) analysis, rather than a modelled estimate of cost-effectiveness in terms of life-years gained. As such, it cannot substitute for a conventional cost-effectiveness analysis using final health outcomes. In its current form, the manuscript risks misinterpretation, as it may give the impression that the reported ICERs can be directly compared to life-year-based thresholds, which is not methodologically appropriate.

To address this issue, the authors should:

a) Either clearly reframe the analysis as a cost-consequence or intermediate-outcome economic evaluation and avoid direct comparison with life-year-based thresholds;

b) Or extend the analysis to estimate downstream health outcomes (e.g., life-years or DALYs gained), allowing ICERs to be expressed in comparable units.

At a minimum, if the current approach is retained, the manuscript should:

a) Explicitly state that direct comparison to life-year-based thresholds is not valid;

b) Clearly describe the "life-years required" results as a threshold-based (exploratory) analysis, not a modelled ICER;

c) Avoid drawing definitive cost-effectiveness conclusions based on these thresholds.

Given that this issue affects the core interpretation of the results and has been raised in previous rounds, I consider this a major concern that should be resolved prior to acceptance.

Minor comments

2) The deterministic sensitivity analysis (tornado diagram) appears to be presented in terms of cost per test, rather than ICERs or net monetary benefit (NMB). As cost-effectiveness decisions are based on incremental costs and effects relative to a willingness-to-pay threshold, presenting results in terms of cost per test limits interpretability and does not align with standard economic evaluation practice. The authors should consider presenting the sensitivity analysis using ICERs or, preferably, NMB, to better reflect the decision-making framework.

3) Figure 5 presents the required life-years per early ART initiation as a function of willingness-to-pay thresholds.The manuscript does not clearly specify the underlying calculation. To ensure transparency and reproducibility, the authors should explicitly report the formula linking incremental cost per early ART initiation to the implied life-years required (e.g., ΔC/λ), clarify whether posterior means or draws were used and describe how uncertainty intervals were derived (e.g., from posterior distributions).

[LINK]

---

## [Editor Report · Decision Letter 5]

10 Apr 2026

Dear Dr Elsbernd,

On behalf of my colleagues and the Academic Editor, Marie-Louise Newell, I am pleased to inform you that we have agreed to publish your manuscript "Point-of-care early infant HIV diagnosis at birth in a pragmatic cluster-randomized trial in Mozambique and Tanzania: a comparative cost and cost-effectiveness study" (PMEDICINE-D-25-00626R5) in PLOS Medicine.

**Please confirm that your manuscript contains in the supporting data the minimal dataset required to reach your conclusions (i.e. The values behind the means, standard deviations and other measures reported; the values used to build graphs; the points extracted from images for analysis). Please include an additional Excel file in the SI to satisfy this journal requirement, if necessary.

PRESS

Sincerely,

Alison Farrell, Ph.D.

Senior Editor

PLOS Medicine